# Soft, tough, and fast polyacrylate dielectric elastomer for non-magnetic motor

Li-Juan Yin [1], Yu Zhao[2], Jing Zhu[1], Minhao Yang [3], Huichan Zhao [4✉], Jia-Yao Pei [1], Shao-Long Zhong[1] & Zhi-Min Dang [1,2✉]

Dielectric elastomer actuators (DEAs) with large electrically-actuated strain can build light-weight and flexible non-magnetic motors. However, dielectric elastomers commonly used in the field of soft actuation suffer from high stiffness, low strength, and high driving field, severely limiting the DEA's actuating performance. Here we design a new polyacrylate dielectric elastomer with optimized crosslinking network by rationally employing the difunctional macromolecular crosslinking agent. The proposed elastomer simultaneously possesses desirable modulus (~0.073 MPa), high toughness (elongation ~2400%), low mechanical loss ($\tan \delta_m = 0.21$@1 Hz, 20 °C), and satisfactory dielectric properties ($\varepsilon_r = 5.75$, $\tan \delta_e = 0.0019$ @1 kHz), and accordingly, large actuation strain (118% @ 70 MV m$^{-1}$), high energy density (0.24 MJ m$^{-3}$ @ 70 MV m$^{-1}$), and rapid response (bandwidth above 100 Hz). Compared with VHB$^{TM}$ 4910, the non-magnetic motor made of our elastomer presents 15 times higher rotation speed. These findings offer a strategy to fabricate high-performance dielectric elastomers for soft actuators.

[1] State Key Laboratory of Power System, Department of Electrical Engineering, Tsinghua University, Beijing, China. [2] School of Electrical Engineering, Zhengzhou University, Zhengzhou, China. [3] Institute of Energy Power Innovation, North China Electric Power University, Beijing, China. [4] Department of Mechanical Engineering, Tsinghua University, Beijing, China. ✉email: zhaohuichan@tsinghua.edu.cn; dangzm@tsinghua.edu.cn

As the milestone equipment of the Second Industrial Revolution, electromagnetic motors have greatly motivated the advance of technology and society. Up to now, the intrinsic features of electromagnetic motors like low specific power, bulky frame, rigid structure, and complicated control system, cannot meet the requirement of light weight, high flexibility, and easy control for soft micromotors, which have a wide range of applications in flexible electronics[1,2], soft robotics[3,4], biomedical implantation[5], and aerospace[6,7]. Recently, soft actuators made of various types of electroactive polymers (EAPs) with electric-field-induced deformation have received considerable attention in the aforementioned fields[8–13]. Different from the actuators with ferroelectric polymers[11,13], dielectric elastomer actuator (DEAs), an electrically stimulus-responsive actuator, show great potential for the large strain and superior flexibility[8,12,14,15]. A large variety of commercial elastomers, including silicone rubbers[9,16–18], polyurethanes[19], and acrylic elastomers (e.g., VHB$^{TM}$ 4910 from 3 M)[8,20], have been widely used to assemble DEAs. Among them, acrylic-based elastomers outperform others for composing non-magnetic motors because of their high dielectric constant (~4.4@1 kHz), large area strain (>380%), and high energy density (3.4 MJ m$^{-3}$)[8]. However, commercial VHB$^{TM}$ acrylic elastomers usually require high driving electric field ($E$) (>80 MV m$^{-1}$)[20–22] due to their inherently high stiffness (Young's modulus around 0.2~1.0 MPa)[23], and exhibit slow response speed (bandwidth usually below 10 Hz)[24] and severe viscoelastic creep[8,24], and high mechanical loss due to viscoelastic character (tan $\delta_m$~0.5)[23].

Numerous efforts have been made to reduce the driving $E$ and increase the response speed of acrylic-based elastomer[25,26]. According to the Maxwell stress, the actuation strain ($S_z$)[8] in the thickness direction can be defined by $S_z = -\frac{\varepsilon_0 \varepsilon_r E^2}{Y}$, where $\varepsilon_0$, $\varepsilon_r$, $Y$ and $E$ are the permittivity of vacuum, dielectric constant of elastomer, Young's modulus of elastomer, and applied electric field, respectively. Thus, the driving $E$ for a certain actuation strain can be reduced by enhancing the actuation sensitivity ($\beta$, defined as $\frac{\varepsilon_r}{Y}$), which is usually achieved by either increasing the $\varepsilon_r$ or decreasing the $Y$ of the elastomer[10,25,27,28]. A common strategy to increase the $\varepsilon_r$ of elastomer used is to incorporate fillers with high dielectric constant into the elastomer matrix[27,29–31]. However, a high concentration of fillers is a requisite for achieving a considerable increase of $\varepsilon_r$, yet it, in turn, makes the elastomer stiff[29] and usually causes premature electrical breakdown[31]. Adding plasticizers with short molecular chains into the elastomer matrix is a typical way to decrease the $Y$ of the elastomers[26,32]. Nevertheless, the poor compatibility between the guest and host may lead to the severe leakage of plasticizers under a harsh or multi-cycled environment[32]. As for the response speed, it can be improved by decreasing the hysteresis of elastomer stemming from an amount of inter-molecular interaction[22], yet this method is usually accompanied by the decrease of $\varepsilon_r$[17]. In a word, most strategies with an effort of contributing to the improvement of one performance (i.e., low modulus, high dielectric constant, low mechanical loss) for acrylic elastomers, may cause the degradation of the others. The design and preparation of high-quality dielectric elastomers remain a challenging problem for the mass applications of DEAs.

Here, we report an advanced polyacrylate dielectric elastomer with low Young's modulus (~0.073 MPa), high toughness (elongation ~2400%) and low mechanical loss (tan $\delta_m$ = 0.21@1 Hz, 20 °C), satisfactory dielectric properties ($\varepsilon_r$ = 5.75, tan $\delta_e$ = 0.0019 @1 kHz) by optimizing its crosslinking network. Different from conventional small-molecular crosslinkers, we employ a large-molecular-weight urethane acrylate compound with flexible polyether diol and aliphatic diisocyanate segments[26] as crosslinker to react with $n$-butyl acrylate ($n$BA) as the monomer to minimize the functionality of crosslinking points. The flexible polyether diol segment of macromolecular crosslinkers can act as "lubricant" and alleviates the strong dipole-dipole interaction between adjacent side groups of poly($n$BA), which leads to the achievement of low elastic modulus (soft) and low mechanical loss (fast). Matching the average molecular weight among crosslinking points ($\bar{M}_c$) of elastomers contributes to eliminating stress concentration and delaying the failure of elastomers (tough). Besides, a certain amount of uncrosslinked chains with strong oriented polarization will increase the dielectric constant of as-synthesized elastomers. Finally, we fabricate a rotational motor apparatus to verify its optimized performance of actuation.

## Results and discussion

**Polymer network and mechanical properties.** The crosslinked network structure of elastomer has a significant influence on the intrinsic mechanical and electrical properties of dielectric elastomer, accordingly affecting its actuation behavior achieved by the conversion of electrical to mechanical energy. The polymer network is usually established by the chemical bonding between the polymer main chains and crosslinking agents with multiple reactive groups or the physical entanglement of macromolecular chains. In this contribution, we innovatively develop a strategy for the optimization of crosslinked polymer networks through modulating the average molecular weight (dimension) of crosslinking agents. Thus, we reduce the functionality of crosslinking points from 4 (Fig. 1aii) to 3 (Fig. 1aiv and avi) and thereby increase isotropic homogeneity due to more circle-like meshes (quadrilateral network for BA-S, shown in Fig. 1aii; hexagon network for BA-M and BA-L, shown in Fig. 1aiv; equal hexagon network for BAC2 shown in Fig. 1avi). Here, the monomer used for the polymerization of acrylic elastomer and the corresponding crosslinking agent were theoretically selected to acquire a uniform crosslinked polymer network (see "Methods" for selection of number average molecular weight). After that, the monomer of $n$-butyl acrylate ($n$BA) was adopted for the fabrication of acrylic elastomer due to the good flexibility, appropriate intermolecular interaction, and broad working temperature range of $n$-butyl acrylate homopolymer (Supplementary Fig. 1). In addition, the oligomer with polyether as repeat units was chosen for serving as the crosslinking agent due to its good flexibility and solubility in $n$BA monomer matrix (see "Methods" for network design and Supplementary Fig. 2).

The average molecule weight of crosslinker should be in the range of $10^4$ to $10^5$ g mol$^{-1}$, which matches the range of average molecular weight ($\bar{M}_c$) between crosslinking points of polymer network (see "Methods" for selection of number average molecular weight). Considering the analysis discussed above, CN9021NS ($\bar{M}_n = 28000$ g mol$^{-1}$), a difunctional urethane acrylate compound composed of a flexible polyether diol segment and an aliphatic diisocyanate segment, was chosen as macromolecular crosslinker for the construction of $n$-butyl acrylic-based elastomer network (BAC). Herein, the dimension of crosslinkers match well with that of network mesh constructed by themselves. For comparison, $n$-butyl acrylate homopolymer (BA-S, BA-M and BA-L), crosslinked by the equimolar (taking BAC2 as reference) polyethylene glycol diacrylate (a small-molecular crosslinker, $\bar{M}_n = 575$ g mol$^{-1}$), CN9893NS (a medium molecular crosslinker, $\bar{M}_n = 1600$ g mol$^{-1}$) and CN9014NS (a large molecular crosslinker, $\bar{M}_n = 6800$ g mol$^{-1}$), respectively, was also prepared through a similar polymerization process.

The terminal vinyl groups (–CH=CH2) of CN9021NS were chemically bonded into the adjacent polymer main chains during the photo-polymerization process and a structure-controlled and uniform hybrid network was subsequently obtained (Fig. 1avi).

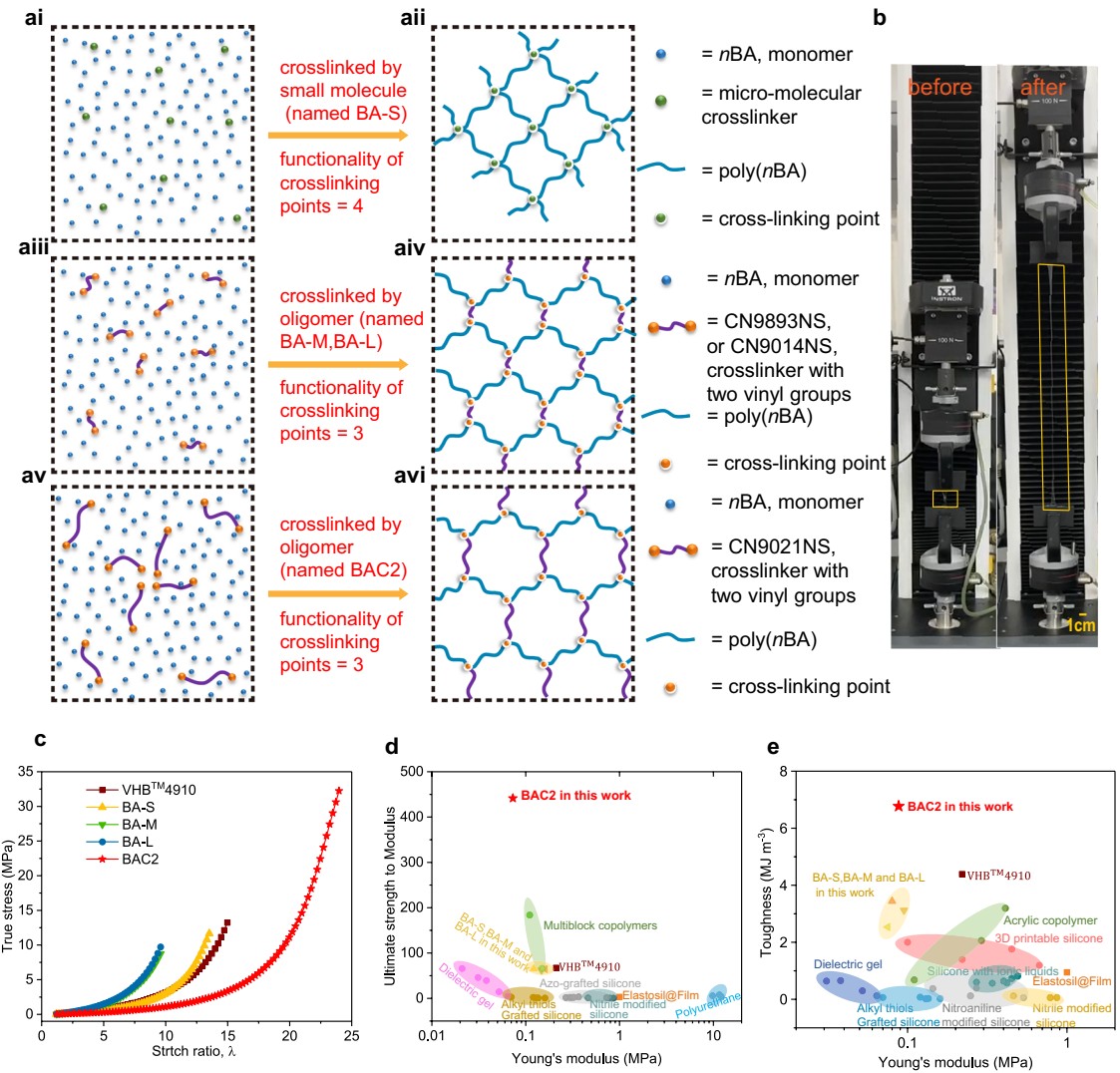

**Fig. 1 Performance of uniform hybrid polymer network. a** The precursor (**ai**, **aiii**, **av**) consists of *n*-butyl acrylate (*n*BA) as monomers and small-molecules (**ai**) or oligomers with two vinyl groups (–CH=CH₂) (**aiii** and **av**) as crosslinkers. When exposed to ultraviolet light (UV-light), the precursor is cured to a flexible network (**aii**, named BA-S; **aiv**, named BA-M or BA-L; **avi**, named BAC2). Distribution of molecular weight for those oligomers have been tested by gel permeation chromatography (GPC) and shown in Supplementary Table 1. **b** Comparison of BAC2 before (left) and after (right) uniaxial stretched. **c** The stress–strain curves of commercial VHB™4910 and BA-S, BA-M, BA-L, BAC2 at a stretch rate of 200 mm min⁻¹. **d** Comparison of ultimate strength to modulus among VHB™ 4910, BA-S, BA-M, BA-L, BAC2, dielectric gel[37], multiblock copolymers[33], azo-grafted silicone[28], nitroaniline modified silicone[18], polyurethane[19], alkyl thiols grafted silicone[16] and commercial Elastosil@Film[16]. **e**, Toughness plotted against Young's modulus for VHB™ 4910, BA-S, BA-M, BA-L, BAC2, dielectric gel[37], silicones with ionic liquids[17], acrylic copolymer[25], nitroaniline modified silicone[18], nitrile modified silicone[39], alkyl thiols grafted silicone[16], 3D printable silicone[40], and commercial Elastosil@Film[16] (Supplementary Discussion).

Accordingly, as shown in Fig. 1b, c, the BAC2 specimen exhibits great stretchability (elongation ~2400%), much higher than that of BA-S, BA-M, and BA-L because of lower crosslinking density and more-balanced network structure. Meanwhile, all the samples (BA-S, BA-M, BA-L, and BAC2), own a low elastic modulus (0.073 MPa~0.161 MPa, summarized in Supplementary Table 1, Supplementary Fig. 3), which is attributed to the high flexibility of poly(*n*-butyl acrylate) chains and reduced degree of crosslinking. According to Young's modulus, the average molecular weight among crosslinking points ($\bar{M}_c$) was estimated for all samples synthesized here, and, as expected, $\bar{M}_n$ of CN9021NS matches best with the $\bar{M}_c$ of the elastomer crosslinked by it.

Generally, it is a huge challenge to simultaneously improve the softness and toughness of elastomer and such a paradox is solved in this contribution through the optimization of crosslinked network. The BAC2 sample not only exhibits a low Young's modulus and a high elongation but also displays a high toughness (6.77 MJ m⁻³) and an ultrahigh ultimate strength (32.2 MPa), which, to the best of our knowledge, surpasses that of the most advanced dielectric elastomers[33] (Fig. 1d, e, and Supplementary Fig. 4).

**Influence of chemically uncrosslinked chains.** A crosslinked polymer network is usually composed of the crosslinked chains and an appropriate amount of dissociative chains that are not involved in the construction of a network (Fig. 2a). The proportion of dissociative chains in a network crosslinked by macromolecular agents is probably increased due to the "cage effect" and steric hindrance of macromolecular chains, which can significantly weaken the reactive probability of terminal groups of macromolecular crosslinking agents (CN9021NS). Thus, the crosslinking efficiency and density will be subsequently

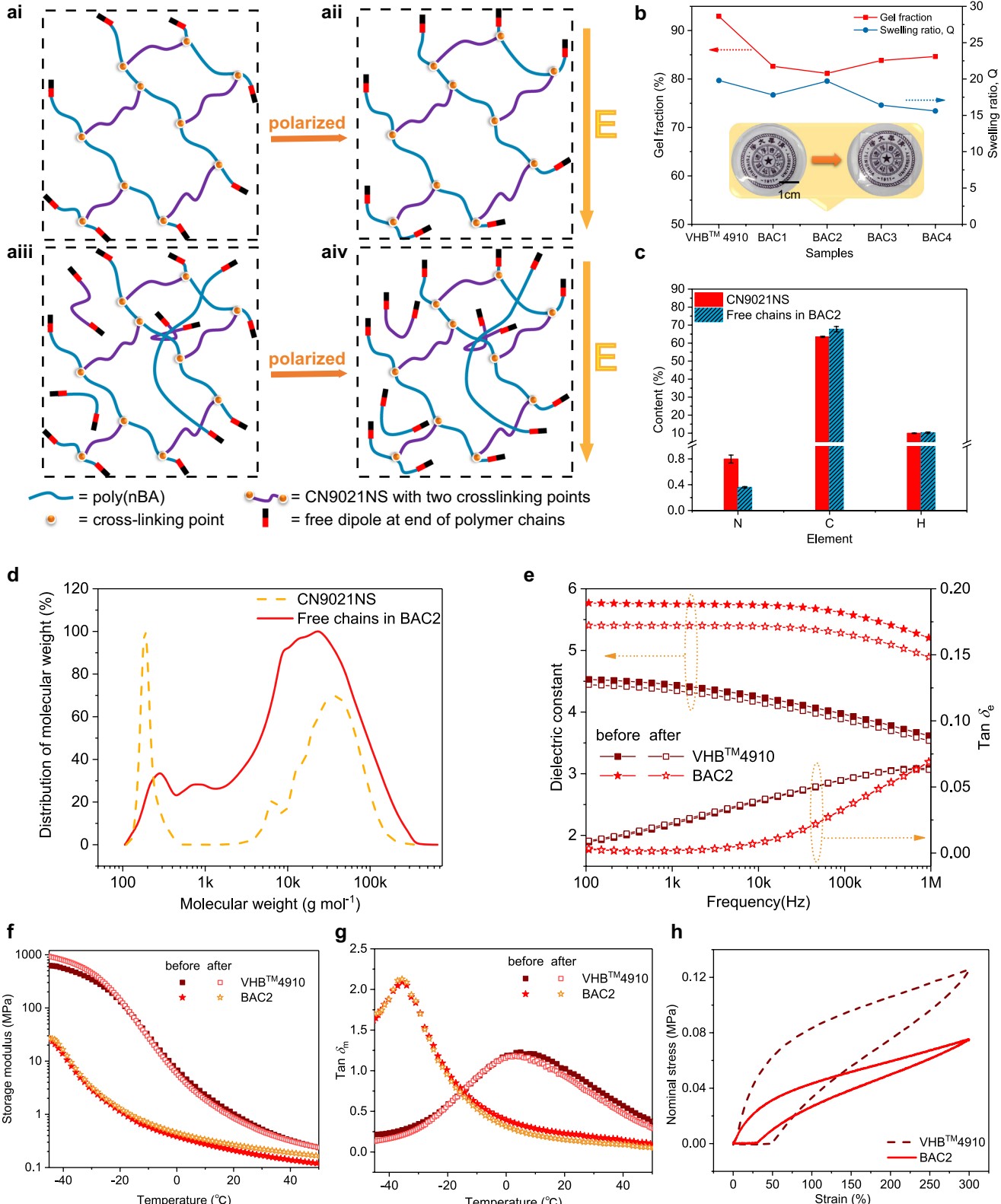

**Fig. 2 Influence of chemically uncrosslinked chains on dielectric and mechanical properties. a** Schematic comparison of the polarization process of acrylic elastomer without uncrosslinked chains (**ai**, **aii**) and BAC2 sample with a considerable amount of uncrosslinked chains (**aiii**, **aiv**). **b** Gel fraction and swelling ratio of VHB™4910 and synthesized elastomers. The insets show the optical images of BAC2 sample before and after swollen. **c** Elemental analysis comparison between CN9021NS and free chains of BAC2 after swollen. At least three samples were tested for the calculation of average value and standard deviation. **d** Molecular weight distribution of CN9021NS and free chains of BAC2 after swollen. **e** Frequency dependence of dielectric constant and dissipation factor (Tan $\delta_e$) of VHB™4910 and BAC2 before and after swollen in the range of $10^2$–$10^6$ Hz. **f, g** Storage modulus (**f**) and mechanical loss (Tan $\delta_m$) (**g**) as a function of temperature of VHB™4910 and BAC2 before and after swollen. **h** The cyclic stress–strain curves of VHB™4910 and BAC2 at a stretch rate of 100 mm min$^{-1}$. The area encircled by the curve indicates the dissipated mechanical energy.

suppressed. Such a hypothesis is verified by the swell experiment and the result indicates that BAC2 sample owns the highest fraction of dissociative component (18%) when compared with VHB™ 4910 and other BAC samples (Fig. 2b). The analysis of the element and molecular weight distribution of CN9021NS and free chains extracted from BAC2 sample after the swelling experiment indicates that the dissociative component is mainly composed of unreacted CN9021NS and uncrosslinked poly(*n*-butyl acrylate) chains (Fig. 2c, d).

We assume that the effect of these free chains within the crosslinked network on the actuation performance of elastomer should not be neglected. The dielectric and mechanical properties of elastomers are of great importance for the actuation performance of dielectric elastomers. Dielectric analysis of VHB™ 4910 and BAC samples was carried out and the result shows that BAC2 sample exhibits the highest dielectric constant value when compared with those of VHB™ 4910 and other BAC samples (Fig. 2e and Supplementary Fig. 5b). Usually, the long flexible polyether segment in CN9021NS may contribute to a decrease of dielectric constant for BAC2 because of its lower dipole density compared with that of poly (*n*-butyl acrylate)[26]. As for the loss part, BAC samples exhibit significant low dielectric loss (tan $\delta_e = 0.0019$ @ 1 kHz) than that of VHB™ 4910 (tan $\delta_e = 0.0229$ @ 1 kHz), and no obvious difference for the loss of BAC series samples has been observed (Fig. 2e and Supplementary Fig. 5b). In order to figure out the contribution of the free chains inside the crosslinked network to the dielectric properties of elastomer, the comparison of the dielectric properties of VHB™ 4910 and BAC2 samples before and after the swelling experiment was performed. Compared with the change of VHB™4910, the dielectric constant of BAC2 sample after swollen displays an obvious decrease from 5.75 for the original one to 5.4 while the loss almost remains unchanged (Fig. 2e).

In addition, the comparison of storage modulus and mechanical loss before and after swollen indicates the existence of these free chains does not raise the negative effect on the mechanical properties of elastomer (Fig. 2f, g).

Different from those conventional strategies implemented by incorporating fillers with high dielectric constant into elastomer matrix or grafting polar groups into elastomer main chains, the approach employed in this contribution apparently eliminates the occurrence of the inevitable deterioration of dielectric loss and mechanical properties of the elastomer. More importantly, these free chains inside the elastomer matrix are helpful to improve the dielectric and mechanical properties of elastomer, and thus ameliorating actuation performance. The contribution of these free chains to the dielectric and mechanical properties of elastomer could be explained from the following two aspects. Firstly, the free chains have more orientation freedom than that of constrained segments in crosslinked network. These free chains not only afford the elastomer strong polarization intensity under external electric field, but also increase the density of dipoles in the elastomer matrix (Fig. 2aiii, aiv). Therefore, the dielectric properties of elastomers with dissociative chains are significantly improved (Fig. 2e). Secondly, the elastic modulus of the elastomer mainly depends on the crosslinked network. Thus, elastomers with these free chains are a little softer since these free chains have not been involved in the construction of the polymer network (Fig. 2f). More importantly, for the mechanical loss obtained from the cyclic stress–strain curve, the loss of BAC2 sample (tan $\delta_m = 0.21$@1 Hz, 20 °C) is much lower than that of VHB™ 4910 (tan $\delta_m = 0.93$@1 Hz, 20 °C) (Fig. 2g, h), which may contribute to the improvements in electro-mechanical conversion efficiency and driving speed of elastomer. On the one hand, *n*BA, the only monomer, only has a medium polar dipole-ester group, and additionally, the ester group connects a butyl at the other end, which serves as lubricants to accelerate polymeric

segmental relaxation. On the other hand, the flexible long aliphatic polyether backbone inside CN9021NS can act as the "spacer" among adjacent polyacrylate chains and the introduced "spacer" will weaken dipole-dipole interaction, and thus reduce mechanical energy loss during the deformation process. The suppressed loss obtained here is essentially important for the reduction of heat production during the cyclic deformation process.

**Static actuation performance**. When dielectric elastomers are used for actuators, actuation sensitivity ($\beta$) is one of the most important parameters that is usually used to evaluate the actuation performance and is defined as the ratio of dielectric constant and Young's modulus. As discussed previously, the dielectric constant of BAC2 sample is of the highest value among those commercial and lab-made samples, whereas its Young's modulus is obviously lower than that of VHB™ 4910. Accordingly, owing to the high $\varepsilon_r$ and low $Y$, the actuation sensitivity of BAC2 elastomer film can reach up to 78.8, which is 3.75 times larger than that of VHB™ 4910 ($\beta = 21$) (Fig. 3a).

In order to clearly evaluate the actuation performance of elastomers, an experimental apparatus was specially designed (Supplementary Fig. 6). Due to the improved actuation sensitivity, BAC2 elastomer exhibits a largest area strain of 18.5% at 15 MV m$^{-1}$ without pre-strain. Nevertheless, the area strain obtained from VHB™ 4910 film at the same electric field is only 4.5% (Fig. 3b). It is worth noting that the actuation electric field is much lower than the electrical strength of samples. For detailed information about experiments and measured data of electric breakdown, please refer to Supplementary Fig. 7. When the films were pre-stretched equiaxially to four times their original diameter, the area strain achieved in BAC2 elastomer film reaches up to 118% at 70 MV m$^{-1}$, which is almost 3.5 times larger than that of commercial one at the same electric field (Fig. 3c). In addition, at the nominal driving electric field of 70 MV m$^{-1}$ (4.375 kV), the energy density of BAC2 is estimated to 0.242 MJ m$^{-3}$ according to the theoretical model set out by Pei et al.[8], while, under the same conditions, that of VHB™ 4910 is only 0.042 MJ m$^{-3}$ (Supplementary Fig. 8). Such a decent property can be attributed to higher dielectric constant and larger actuation strain.

**Dynamic response and cyclic actuation**. It is well known that the application of pre-strain on the elastomer film can not only suppress electromechanical instability, but also enhance electrical breakdown strength, consequently leading to the increase of maximum actuation strain at a higher electric field. In addition, pre-strain can also be used to alleviate the viscoelasticity of poly-acrylate elastomers[34], which is regarded as the main issue for obtaining fast response speed and stabilizing strain under the applied electric field. As shown in Fig. 3d, the VHB™ 4910 elastomer needs almost 5 min to reach 90% of its final strain, whereas BAC2 elastomer only needs 35.2 s to reach the same relative strain value (90%). In addition to the fast response speed, the area strain of BAC2 elastomer is almost invariable after the application of applied electric field. On the contrary, the commercial one displays a severe creep behavior (Fig. 3d and Supplementary Fig. 9).

To further characterize the dynamic behavior of elastomers, frequency response analysis was performed on VHB™4910 and BAC2, both of the identical geometry adopted to characterize actuation property without pre-strain. A sinusoidal excitation waveform with amplitude of 5 kV and frequency ranging from 1–100 Hz is used to drive the films and frequency dependence of normalized actuated area strain is plotted in Fig. 3e. Owe to

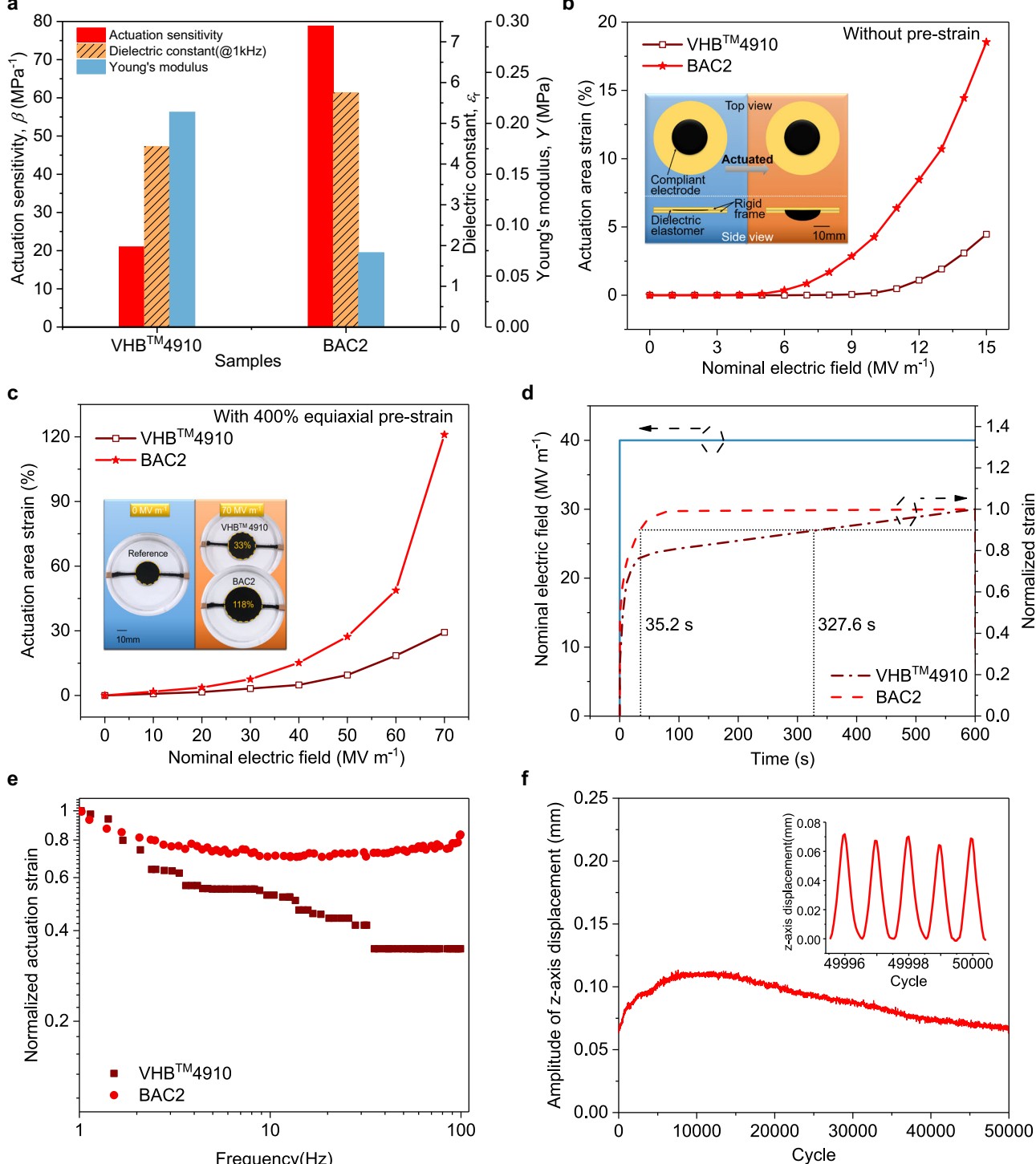

**Fig. 3 Actuation performance. a** Comparison of dielectric constant ($\varepsilon_r$) measured at 1 kHz, Young's modulus ($Y$), and actuation sensitivity ($\beta$) between VHB™4910 and BAC2 sample. **b** Dependence of actuation area strain on the nominal electric field for the films of VHB™4910 and BAC2 elastomers without pre-strain. The nominal electric field is defined as voltage on films divided by initial thickness before electrically actuated. The inset shows the schematic diagrams illustrating the electro-actuation deformation process from the view of top and side directions. **c** Dependence of actuation area strain on the nominal electric field for the films of VHB™ 4910 and BAC2 elastomers with 400% equiaxial pre-strain. The inset displays the optical images of VHB™ 4910 and BAC2 films before and after actuated. **d** Time-dependent behavior of the actuated area strain of VHB™ 4910 and BAC2 films with 400% equiaxial pre-strain at 40 MV m⁻¹. Normalized strain denotes the time-varying strain divided by strain at 600 s. **e** Frequency response of samples to large drive signals (5 kV) in the range of 1–100 Hz. The data were normalized to 1 at 1 Hz. **f** Cyclic actuation test of 50,000 cycles was performed on BAC2 film at electric field of 5 kV and excitation frequency of 5 Hz. The inset demonstrated z-axis displacement response in the last five cycles.

suppressed mechanical loss (Fig. 2g, h), frequency response of BAC2 is nearly flat in the 1–100 Hz frequency range. Because the starting of resonance emerges at 100 Hz, resonance peak is reckoned to locate out of measurement range, which may need more attention in future. Rather, with the increase of frequency, actuation response of VHB$^{TM}$4910 declines sharply, which, as a consequence, greatly limits it to applications below a few Hz. Then, cyclic actuation test of BAC2 was performed successfully for 50,000 cycles based on the identical geometry and setup used above except at a fixed frequency of 5 Hz. z-axial displacement has undergone a mild growth firstly, which may be ascribed to its own gravity, and then fallen back slowly to the level at first several cycles and remain stable (Fig. 3f). The large area strain, rapid electromechanical response and satisfactory anti-fatigue property make BAC2 elastomer film as a promising candidate in the actuation fields with the requirements of low electric field operation and potential broad frequency output.

**Flexible non-magnetic motor.** When a film is divided into four parts, and electrically driven clockwise or anticlockwise around the rotor settled in the center of the film, the rotor will rotate in the opposite direction. In this process, electromechanical energy conversion is achieved, and thus, this apparatus is called soft motor[20–22]. The motor will rotate when electric field for actuation exceeds a threshold (the minimum driving electric field) due to the space interval between rotator and inner edge of annular film as well as gear fraction.

Considering the superior dielectric, mechanical, and actuation performance of BAC2 film, the motor made of BAC2 film may rotate at a lower electric field and in a broader frequency range compared with the one using VHB$^{TM}$4910 film. Figure 4a displays a complete rotation process for the BAC2 based motor. As illustrated in Fig. 4b, the rotation speed exhibits a monotonous increase with driving frequency under a fixed electric field. Besides, the maximum rotation speed increases obviously when the electric field is enhanced. The minimum driving electric field is a vitally important parameter in evaluating the actuation performance of elastomer film. The minimum driving electric field is significantly reduced from 48 MV m$^{-1}$ for VHB$^{TM}$ 4910 based motor to 32 MV m$^{-1}$ for BAC2 based one, which is attributed to the improved actuation sensitivity. In addition, owing to the fast response, BAC2 based motor could be triggered to rotate at a higher frequency than VHB$^{TM}$ 4910 based motor (Fig. 4b). At the same driving electric field (48 MV m$^{-1}$), the maximum rotation rate of BAC2 based motor is 0.72 r s$^{-1}$, which is 15 times larger than that of VHB$^{TM}$ 4910 based motor. Furthermore, the rotation rate of the motor could be further increased to 2.86 r s$^{-1}$ when equipped with transmission gears (Fig. 4c, Supplementary Movies 1, 2, and 3).

The output torque and power of motors mainly depend on the Maxwell pressure, strain, and mechanical loss of elastomers. The Maxwell pressure on the elastomer film scales linearly with dielectric constant and quadratically with the applied electric field while the strain is defined as the ratio of Maxwell pressure and Young's modulus[28]. Due to the significant improvement of dielectric constant, mechanical loss, and actuation performances as discussed above, the output torque and power of BAC2 based motor are distinctly improved (Fig. 4d). The output torque and power are 6 times and 18 times larger than those of VHB$^{TM}$4910 based motor, respectively. The fabrication process and characterization of these soft motors are displayed in the "Methods" section in detail.

Limited by the experimental equipment and dimension of motors, the time constant for charging is at the magnitude of 10$^{-2}$ s. Thus, with the rise of driving frequency, the maximum

voltage on film descends severely. Simulation from MATLAB/Simulink reveals that the output performance of VHB$^{TM}$4910 based motor is bottlenecked by elastomer itself. However, the output performance of BAC2 based motor is not only related to the elastomer but also restricted by the experiment platform's power (see Supplementary Discussion, and Supplementary Figs. 10 and 11). Herein, employing a high voltage and high-power amplifier with large output current could improve the actuation frequency of BAC2. As a result, several key issues associated with high actuation voltage, slow response speed, and severe viscoelastic loss have been resolved in this contribution. The approach developed here through the rational optimization of crosslinked elastomer networks provides an effective strategy to tackle these challenges fundamentally.

Finally, we have demonstrated an innovative strategy to improve the actuation performance of dielectric elastomers by optimizing crosslinking network. These performances could be mainly attributed to the flexible long-chain structure of crosslinking agents and the existence of dissociative chains inside the network. This strategy readily enables elastomers with extraordinary soft, ultra-high toughness, satisfactory dielectric properties, large actuation, and fast response. Large actuation and high energy density under low electric field is promising for the development of soft actuators in low-voltage driving fields. The principle for the selection of elastomer component and strategy employed here provides a different insight for high-performance dielectric elastomers and actuators.

## Methods

**Network design.** UV-polymerization has been widely used for the synthesis of polymers due to its simplicity, controllability, and low polymerization temperature. Thus, UV-polymerization was selected in this work to synthesize the acrylic-based dielectric elastomer. Firstly, a quantitative structure-property relationship (QSPR)[35] approach was used to theoretically select the optimum monomer from 11 commonly used acrylate monomers. Some critical physical parameters of these acrylate polymers including molar stiffness function (MSF), cohesive energy (CE), and glass transition temperature ($T_g$), which represent the flexibility, intermolecular interaction energy, and minimum working temperature, respectively, were calculated by this QSPR method (Supplementary Fig. 1). The obtained results indicated that n-butyl acrylate (nBA) monomer exhibited the optimum comprehensive properties for the synthesis of acrylic-based dielectric elastomer.

Moreover, the MSF and $T_g$ of 18 commonly used flexible long-chain oligomers with polyester, polyether, and silicone as repeat units as well as the compatibility between these oligomers and BA monomers were calculated (Supplementary Fig. 2). Considering the features of flexibility and solubility, the oligomers with polyether as repeat units were recommended for the optimization of the network of n-butyl acrylic-based elastomers. In addition, the average molecular weight of oligomers should be in the range of 10$^4$ g mol$^{-1}$ to 10$^5$ g mol$^{-1}$ since a structurally uniform polymer network can only be achieved in this molecular weight range (see "Methods" for selection of number average molecular weight). According to the calculated results and analysis discussed above, CN9021NS ($\bar{M}_n$ = 28,000 g mol$^{-1}$), a difunctional urethane acrylate compound composed of a flexible polyether diol segment and an aliphatic diisocyanate segment, was chosen for elastomer network optimization (see "Methods" for selection of number average molecular weight). For comparison, n-butyl acrylate homopolymer (BA-S, BA-M, and BA-L), crosslinked by the equimolar (taking BAC2 as reference) polyethylene glycol diacrylate (a small-molecular crosslinker, $\bar{M}_n$ = 575 g mol$^{-1}$), CN9893NS (a medium molecular crosslinker, $\bar{M}_n$ = 1600 g mol$^{-1}$) and CN9014NS (a large molecular crosslinker, $\bar{M}_n$ = 6800 g mol$^{-1}$), respectively, was also prepared through a similar polymerization process.

**Selection of number average molecular weight.** According to the previously reported studies, crosslinking density of dielectric elastomers should be in the range of 10$^{-5}$ mol cm$^{-3}$ to 10$^{-4}$ mol cm$^{-3}$[36]. The density of elastomers is usually around 1 g cm$^{-3}$. Thus, the range of the crosslinking density could be derived as 10$^{-5}$–10$^{-4}$ mol g$^{-1}$. Afterwards, the average molecular molar weight between adjacent crosslinking points should be in the range of 10$^4$–10$^5$ g mol$^{-1}$.

Uniform distribution of crosslinking points is helpful to alleviate or eliminate the stress concentration inside the elastomer network, which is usually regarded as a key issue that leads to the early failure of the elastomer. Therefore, a difunctional oligomer with an average molecular molar weight in the range of 10$^4$–10$^5$ g mol$^{-1}$ is desirable for the crosslinking of elastomer. Based on this principle, CN9021NS, a

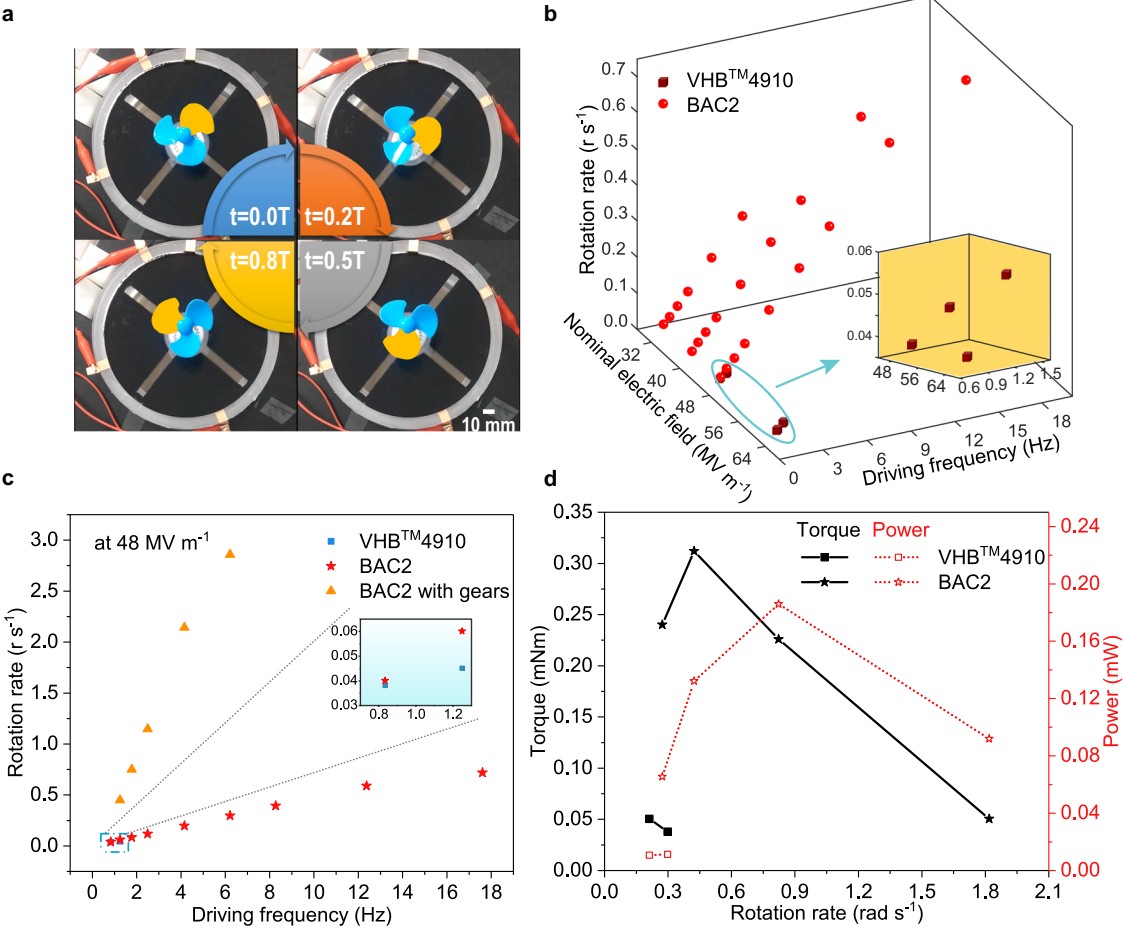

**Fig. 4 Mechanical output performance of soft non-magnetic motors made of dielectric elastomer films. a** Snapshots from the Supplementary Video 1 of a spinning non-magnetic motor displaying a complete rotation cycle, where $t$ and $T$ denote time and rotation period, respectively. **b** Dependence of the rotation rate on electric field and driving frequency for the motors made of VHB[TM] 4910 and BAC2 films. The inset is a magnification of the data indicated by the arrow, which represents the electric field and frequency-dependent behavior of the rotation rate for VHB[TM]4910 based motor. **c** Dependence of the rotation rate on the driving frequency for the VHB[TM] 4910 and BAC2 based motors as well as BAC2 based motor with transmission gears at 48 MV m[−1]. **d** Comparison of the output torque and power of VHB[TM] 4910 and BAC2 based motors.

kind of macromolecular crosslinking agent, was specifically selected and its mean relative molecular weight was characterized by gel permeation chromatography (GPC) in tetrahydrofuran solvent (Shimadzu LC20/RID-20). Further, micro-molecular crosslinker and other difunctional oligomers, whose mean relative molecular weight are below this range, were selected to cure the elastomers for comparison. Estimated average molecular weight among crosslinking points and mean relative molecular weight of those oligomers were summarized in Supplementary Table 1.

**Preparation**. The dielectric elastomer was synthesized via a UV curing method. *n*-butyl acrylate (BA, 99.5%) from J&K Scientific Ltd was selected as monomer. *N,N'*-Ethylenebisacrylamide from Tokyo Chemical Industry Co., Ltd, and CN9021NS, CN9893NS, CN9014NS from Sartomer Company were used as crosslinking agents. 2-hydroxy-2-methylpropiophenone from Shanghai Macklin Biochemical Co., Ltd was used as photo-initiator. For BAC series, formulations composed of monomer, CN9021NS as crosslinker and photo-initiator (as shown in Supplementary Table 2) were mixed thoroughly by vortex oscillator to form a homogeneous precursor solution. For BA-S, BA-M, and BA-L, contents of monomer, crosslinker, and photo-initiator were identical mole ratio to those for BAC2, and preparation process of precursor solution was similar to that of BAC series.

A custom-built vessel composed of a silicone spacer sandwiched by release films and glass plates in a proper order was fabricated to create an oxygen-free condition. The precursor solution was injected into that vessel[37], which was subsequently followed by a vacuum degassing process for 15 min. After that, it was cured in a UV curing reaction chamber with a UV intensity of 2.5 W cm[−2] for 3 min. Finally, the elastomer was placed into vacuum oven at 40 °C for 24 h to remove the remnant monomer. The film thickness can be easily adjusted by tuning the thickness of the silicone spacer.

**Gel experiment and light transmittance test**. All the samples with a thickness of 1 mm were cut into 10 mm × 10 mm squares and their original mass was subsequently measured. Then, they were immersed in the tetrahydrofuran solvent for a week. And the dimensions after swollen were measured to obtain the swelling ratio, which is defined as the cubic ratio of the length of a swollen elastomer to its original length. After drying the samples in vacuum oven at 40 °C for 24 h, the final mass was measured to obtain the gel fraction[9]. Light transmittance test with the wavelength ranging from 380 to 800 nm for the films with a thickness of 1 mm was conducted by Lambda™ 35 UV/VIS Spectrometer from PerkinElmer. Since each formulation exhibits high transparency, BAC2 was taken as a representative sample.

**Elemental analysis**. Firstly, the uncrosslinked chains extracted from BAC2 sample after swollen in the tetrahydrofuran solvent were dried in vacuum oven at 60 °C for at least 24 h to evaporate the solvent completely. Then, CN9021NS and those free chains in BAC2 were measured on PERKIN ELMER CE-440.

**Dielectric properties**. Dielectric properties in the range of 100 Hz to 1 MHz were characterized by a Broadband Dielectric Spectroscopy (Novocontrol Technologies GmbH & Co. KG). The thickness of the samples used for dielectric characterization was 1 mm. The median was selected as a measured value among at least three specimens.

**Mechanical properties**. The measurement was conducted according to the standard of ISO 37. To eliminate the effect of thickness, all samples were in a fixed thickness of 1 mm. After that, samples were cut into a dumbbell shape with an effective area of 2 mm width by 12 mm length and stretched at the rate of 200 mm min[−1] until fracture to obtain stress vs strain curves and Young's moduli were defined as the slopes of tangent lines at 5% strain. Finally, samples were cut into a

narrow strip of 10 mm width by 50 mm length (the gauge length was 30 mm) and stretched to triple at the rate of 100 mm min$^{-1}$ to obtain stress vs strain loops. Each formulation was tested at least three times and the median was selected. Dynamic mechanical analysis (DMA) was carried out on a TA Q800 dynamic mechanical analyzer with a frequency of 1 Hz, <2% strain, and a temperature range from $-45$ to $50\,^{\circ}C$ with a ramping rate of $7\,^{\circ}C\,min^{-1}$.

**Electrical strength test.** A high voltage tester (BDJC-50kV, Beijing, Beiguang) with 25-mm-diameter pillar to plate electrode was used to characterize the electrical strength of elastomers. The elastomers were placed between two electrodes immersed in the silicone oil at room temperature. A DC voltage ramp of 500 V s$^{-1}$ was applied to the electrodes until voltage drops sharply. Ten specimens were tested for VHB$^{TM}$4910 and BAC2 and the characteristic electrical breakdown strength (characteristic $E_b$) could be calculated using a two-parameter Weibull distribution function, $P = 1 - \exp(-(E_b/\alpha)^{\beta})$, where $P$ is the cumulative probability of electric failure, $E_b$ is the measured breakdown strength for each sample, $\alpha$ is the characteristic breakdown strength (characteristic $E_b$) that corresponds to a ~63.2% probability of failure, and $\beta$ is the slope parameter that evaluates the scatter of data. Herein, the characteristic $E_b$ was calculated from a linear fitting using Weibull failure statistics across 10 specimens per sample.

**Actuation properties.** The nominal electric field was calculated by dividing the applied voltage by the initial thickness of elastomer film before electrically actuated. For the films free of pre-strain, the samples with a thickness of 1 mm were fixed on an annular PMMA rigid frame with an inner diameter of 20 mm ($d = 20$ mm). And carbon grease (NyoGel 756 G, Nye Lubricants) was coated onto both sides of the film as the compliant electrodes. The voltage applied on the films was increased at a step of 1 kV until electric breakdown and each strain value was obtained by holding the constant voltage for 1 min. A laser sensor (HL-G105-S-J, Panasonic) was used to measure and record the invagination depth ($h$) in the center of the actuator. Finally, actuation area strain ($S$) is derived from the equation: $S = (2h/d)^2 \times 100\%$ (Supplementary Fig. 6) Each formulation was tested at least three times. As for planar area strain, films were fixed on a PMMA rigid frame with an inner diameter of 120 mm after equiaxially stretched to four times its original diameter (Supplementary Movie 4). And the compliant electrodes with a 25-mm diameter were fabricated by coating carbon grease (NyoGel 756 G, Nye Lubricants) onto both sides of films. The electric field was increased from 0 to 70 MV m$^{-1}$ by a step of 10 MV m$^{-1}$ and each step was held for 30 s. A digital SLR camera was used to record the actuation process. The actuation strain was obtained by a MATLAB script for video processing[38]. At least three samples for the same composition were tested and the median was selected.

**Dynamic response.** The experiment condition in this part was similar to that of the planar area strain characterization except that the electric strength applied on film was 40 MV m$^{-1}$ and held for 10 min. Each formulation was tested at least three times.

**Frequency response.** The samples with a thickness of 1 mm were fixed on an annular PMMA rigid frame with an inner diameter of 20 mm ($d = 20$ mm). And single-wall carbon tube was transferred to both sides of the film as the compliant electrodes. A sinusoidal excitation waveform with peak-to-peak amplitude of 10 V and offset of 5 V was generated by a two-channel function generator (DG4202, Rigol) and then delivered to a high voltage amplifier with a gain of 500 times (AMT-5B20, Matsusada). A laser displacement sensor (LK-g80, Keyence) was used to measure and record the invagination depth ($h$) in the center of the actuator. Finally, actuation area strain ($S$) is derived from the equation: $S = (2h/d)^2 \times 100\%$ (Supplementary Fig. 6). The areal actuation strain was normalized by the strain at 1 Hz.

**Cyclic test.** Test platform and geometry of samples were identical to those used in frequency response testing except a fixed frequency of 5 Hz.

**Fabrication and actuation performance of non-magnetic motor.** After equiaxially stretched to four times its original diameter, films were fixed on a PMMA rigid frame with an inside diameter of 140 mm using a specially designed stretching rig and the films were divided into four parts by electrodes. And carbon grease (NyoGel 756 G, Nye Lubricants) was used for the fabrication of compliant electrodes. In the center, a circular ring with an inside diameter of 40 mm was used to maintain pre-strain and fix gears. Single-chip machine (SCM) was written to control relays on and off. Thus, the frequency of driving voltage can be easily tuned by adapting the SCM script. Rotary motion was recorded by a digital SLR camera, and rotation property was obtained from the video frame-by-frame. The schematic diagram of non-magnetic motor driving system is shown in Supplementary Fig. 10. The motor was mounted with its axis aligned vertically and a string was bound with the rotor and ran over the pully mounted on table to connect a basket[21]. Then, torque was measured by placing a fixed weight in the basket at each driving electric field and frequency. The weight was increased until the motor lost its

synchronization with driving electric field. At last, the output power was calculated by the product of torque and rotation speed.

## Data availability
The authors declare that the data supporting the findings of this study are available within the paper and its Supplementary Information files or from the corresponding authors upon reasonable request.

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

## Acknowledgements

This work was financially supported by the National Natural Science Foundation of China (No. 51937007).

## Author contributions

Z.M.D., L.J.Y., and H.Z. conceived the idea. L.J.Y., Z.M.D., and Y.Z. designed the experiments. J.Z. and S.L.Z. performed quantitative structure-property relationship (QSPR) calculations. L.J.Y., Y.Z., and J.Y.P. carried out the experiments. L.J.Y and J.Y.P. performed the MATLAB simulations. L.J.Y., J.Y.P., and Y.Z. carried out the electrical actuation testing. L.J.Y., Z.M.D., Y.Z., J.Z., J.Y.P., M. Y., S.L.Z., and H.Z. analyzed the data. L.J.Y., Z.M.D., and M.Y. wrote the manuscript. All authors discussed the results and commented on the manuscript.

## Competing interests

The authors declare no competing interests.
