## [Peer Review File · Nature Communications]

Reviewers' comments:

Reviewer #1 (Remarks to the Author):

Dielectric elastomers are a promising form of artificial muscle. In these materials, high fields are applied across thin sheets of these compliant materials. The forces resulting from these high fields lead to deformation of the material - which is compressed in the direction of the applied field, and expands perpendicularly. Pelrine and team at SRI showed that polyacrylate (and in particular the 3M product VHB), showed the highest work density of any of a wide range of elastomers tested. However, the material requires very high actuation voltages, and often suffers from breakdown. Voltages can be lowered by adding fillers, which increase dielectric constant, but this both increases stiffness and reduces breakdown strength, and so typically does not improve actuation. The authors have synthesized and tested a polyacrylate that maintains relatively low elastic modulus, while also having increased dielectric constant. This is achieved by using long chain cross-linkers, which, it is claimed, reduce stress concentrations and enable lower modulus. They also have dangling chains with polarized groups that appear to act to enhance dielectric constant.

Comparisons are made with VHB - which is the standard - but since it is commercially produced, the synthesis conditions, additives and general composition are not known. This makes it difficult to be confident that the authors' claims regarding the underlying mechanisms of improvement, relative to VHB, are justified. The main text would benefit from more analysis gained through comparison with the various versions of the BAC.

In comparing with VHB - how did you decide how high a field to apply? What is the breakdown strength of your material, and how does it compare with VHB?

It is stated that "As shown in Fig. 3d, the VHBTM 267 4910 elastomer needs almost 5 min to reach 90% of its final strain". What is the reason for the low speed in both cases? Is it electrical connection, viscoelastic response or some other effect? How do you know? How much creep is there when the voltage is returned to zero? Can you bring the conclusions of your analysis in the supplementary material, to the main text?

One of the challenges with VHB is its frequency response. While the new material presented here may be a little faster, what are the prospects for getting much higher frequency response?

More perspective would be valuable. For example, I don't see a discussion of work density. This is a very important property. Is it better than in VHB? I also don't see a comparison with actuation in other dielectric elastomer materials, which, for example, can offer much better frequency response. And there is little perspective given on the way forward. How do these materials fit within the theoretical performance criteria set out by Zhigang Suo? It is mentioned that frequency response can be improved - but some speculation on how much, and what this would mean for applications, would be helpful (added to the main text). Also - there is no discussion of breakdown or cycling/failure. Why not? What are the prospects here, compared to VHB and other materials?

Overall I believe this work is making a significant contribution, but it is incomplete, and the authors' reasonable hypotheses on the underlying mechanisms are not well justified in the main text.

Reviewer #2 (Remarks to the Author):

The manuscript reported an optimizing crosslinking strategy to prepare a dielectric elastomer with low loss and outstanding actuation performance. The strategy is clearly presented. The dielectric, mechanical and actuation performance of the elastomers are demonstrated by experiments. The results are solid and interesting, but clear major flaws exist and I cannot recommend its publication in the current form. The following questions should be carefully addressed:

1. The authors claimed the newly synthesized material has low loss, as seen in the title of the paper. However, as shown in Fig. 2h, the hysteresis is still large. The characterization of mechanical loss is too limited and no mechanism was discussed.
2. The authors claimed the newly synthesized material has outstanding actuation performance. This conclusion was drawn based on the experimental results in Fig. 3b, 3c, where the actuation performance was compared with the commonly used DE material VHB. However, the actuation strain of VHB achieved by the authors was too small compared to those in literature, and it was apparently not a fair comparison. To the reviewer's opinion, the actuation performance of the new material is not as good as VHB. See a recent review about the actuation strain of VHB. "Mechanics of dielectric elastomer structures: A review, Extreme Mechanics Letters 2020, 38, 100752".
3. In Fig.3(b-d) and Extended Data Fig.5(b-f), the authors claimed that the optimizing network helps to obtain excellent dielectric and actuation performance. However, in these figures, the performances of BAP are not presented to show how the optimizing crosslinking strategy can improve the dielectric and actuation performance.
4. In Extended Data Table 1, weight ratio of CN9254NS is over 30%, most of which are not reacted. Do the unreacted CN9254NS leak like the plasticizers? Given that there is a large amount of double bonds in the unreacted CN9254NS, do these double bonds make the dielectric elastomer unstable in actuation performance after long-time service?
5. The mechanical tests in the paper seemed not be professional. In the supporting information (Line 86), why 500 is chosen to be the stretch rate for Young's modulus test? Why are the stretch rate for Young's modulus test, fracture test and stress-strain loops not the same (Line86-91 supporting information)? The material should be rate-dependent.
6. In the manuscript, the authors claimed that the BAC2 sample has a high toughness of 6.77J/m^3 , and Fig. 1e shows the diagram of toughness. However, in mechanics test, toughness has a standard definition by measuring a sample with a pre-cut crack and has a unit of J/m^2 .

Reviewer #3 (Remarks to the Author):

This paper reports the design of polyacrylate elastomers for dielectric actuation. The prepared materials are slightly softer, more extendable, and less viscous than commercial VHB elastomers.

Therefore, their actuation properties are slightly better than VHB, which cannot be viewed as revolutionary. Also, the idea behind the pursued materials design strategy needs better justification. Currently, it looks like a trial-and-error approach. What is the major innovation in this paper?

Line 15: Why is 0.088 MPa considered as a desirable Young's modulus? Would a modulus of 0.01 or even 0.001 MPa be more desirable? I guess, it depends on application.

In line 16, 118% should not be considered as a "huge actuation strain". Many DEA systems show larger strains at lower fields (70 MV/um is a relatively high field).

The modulus of 88 kPa corresponds to the lower limit for conventional polymer networks controlled by chain entanglements. Furthermore, enhancement of chain flexibility (by e.g., introducing polyether diol segments in CN9021NS crosslinker) promotes entanglements and therefore raise the lower limit for network modulus. In other words, the design strategy discussed in lines 75-82 is not "helpful to achieving low elastic modulus".

Uncrosslinked chains slightly decrease the modulus, however, they are detrimental for network quality. The achieved "stretchability" of 2300%, i.e. elongation-at-break of 3.3, is lower than a theoretical elongation of ~ 6 , which is expected for a PBA network with the modulus of 0.088 MPa. This suggests non-uniform mesh size distribution.

The idea behind "Selection of number average molecular weight" is unclear. Why does the molecular weight (MW) of crosslinker should match the crosslink density? As discussed above, the crosslink density of conventional linear chain elastomers has a lower bound due to entanglements, which corresponds to the molecular weight of the entanglements strand M_e . For PBA, $M_e \sim 30,000$ g/mol. Is it the reason for the CN9021NS selection? In general, network modulus is determined by crosslink density, whereas elongation-at-break also depends on network uniformity. None of these parameters is directly related to the crosslinker MW. It is possible that longer crosslinkers have much lower polymerization rate constant, which promotes network uniformity. In this case, the authors should study mechanical properties as a function of crosslinker MW. Concurrently, one should conduct NMR studies to monitor incorporation of crosslinker to the network structure. Miscibility of CN9021NS and PBA might be an issue during polymerization reaction. They may phase separate with as the MW increases. In general, CN9021NS and PBA miscibility should be discussed. Gel fraction of the prepared elastomers should be measured. I am surprised about the larger hysteresis in Figure 2h. This suggests poor network quality and significant viscous fraction, which could be due to unreacted species and dangles.

The reported dielectric constant is significantly higher than that of PBA. It is unclear how the uncrosslinked chains "would increase the dielectric constant". Some dipoles at chain ends are mentioned. However, their concentration is relatively low to make significant contribution to dielectric properties.

In line 85, "free volume space of elastomer network" is an odd expression.

Throughout the manuscript: area strain should be replaced with areal strain

Comments from reviewers:

Reviewer #1: Dielectric elastomers are a promising form of artificial muscle. In these materials, high

fields are applied across thin sheets of these compliant materials. The forces resulting from these high fields lead to deformation of the material - which is compressed in the direction of the applied field, and expands perpendicularly. Pelrine and team at SRI showed that polyacrylate (and in particular the 3M product VHB), showed the highest work density of any of a wide range of elastomers tested. However, the material requires very high actuation voltages, and often suffers from breakdown. Voltages can be lowered by adding fillers, which increase dielectric constant, but this both increases stiffness and reduces breakdown strength, and so typically does not improve actuation. The authors have synthesized and tested a polyacrylate that maintains relatively low elastic modulus, while also having increased dielectric constant. This is achieved by using long chain cross-linkers, which, it is claimed, reduce stress concentrations and enable lower modulus. They also have dangling chains with polarized groups that appear to act to enhance dielectric constant.

Our Response: We are grateful for your detailed and positive comments on our manuscript. We have carefully revised the manuscript and extended data according to your valuable suggestions. Please refer to the replies below.

1. Comparisons are made with VHB - which is the standard - but since it is commercially produced, the synthesis conditions, additives and general composition are not known. This makes it difficult to be confident that the authors' claims regarding the underlying mechanisms of improvement, relative to VHB, are justified. The main text would benefit from more analysis gained through comparison with the various versions of the BAC.

Our Response: Thanks for your reasonable suggestion. In order to validate the assumption that long-chain crosslinking agents, whose average molecular weight is on the same order of magnitude of molecular weight of mesh, could facilitate the formation of a more uniform network, accordingly eliminating the inner stress concentration, the work was carried out in two stages.

Firstly, CN9021NS ($\overline{M}_n=28000 \text{ g mol}^{-1}$), a difunctional urethane acrylate compound composed of a flexible polyether diol segment and an aliphatic diisocyanate segment, was chosen as macromolecular crosslinker for the construction of *n*-butyl acrylic-based elastomer network (BAC series). For comparison, *n*-butyl acrylate homopolymer (BA-S, BA-M and BA-L), crosslinked by the equimolar (taking BAC2 as reference) polyethylene glycol diacrylate (a small molecular crosslinker, $\overline{M}_n=575 \text{ g mol}^{-1}$), CN9893NS (a medium-molecular difunctional urethane acrylate crosslinker, $\overline{M}_n=1600 \text{ g mol}^{-1}$) and CN9014NS (a large-molecular difunctional urethane acrylate crosslinker, $\overline{M}_n=6800 \text{ g mol}^{-1}$), respectively, was also prepared through a similar polymerization process. Considering that

chemical reactivity of crosslinkers is very susceptible to their neighboring groups, BA-S, BA-M, and BA-L, whose crosslinker has the same groups bonding to crosslinking points as CN9021NS, have been synthesized. And BAP in Original Manuscript has been renamed as BA-S in Revised Manuscript. Secondly, four formulations with different contents of CN9021NS as the crosslinker have been systematically characterized and demonstrated in **Revised Extended Data**, among which BAC2 exhibits an optimum comprehensive performance including the highest actuation sensitivity and superior ductility. As shown in Fig.1c to Fig.1e, BAC2 further precedes BA-S, BA-M and BA-L in stretchability due to a more uniform crosslinking network, which was achieved through rationally selecting macro-molecular crosslinker. Generally, polymer with low elastic modulus (especially lower than 0.1 MPa, like gels) tends to show poor ductility and terrible ultimate strength, while BAC2 synthesized in this work have all those satisfied properties (Fig.1d and 1e). Moreover, after measuring the true cross-sectional area of specimen, we correct the ultimate strength of BAC2 from 1.34 MPa (nominal stress) to 32 MPa. Such a value, to the best of our knowledge, is the highest value among those soft polymers (*Chem. Eng. J.* 2021, 405, 126634; *Nat. Commun.* 2000, 11, 4000). The BAC2 is 60% lower in Young's modulus (0.073 MPa) but 128% higher in tensile strength (32 MPa) than that of VHBTM4910 (0.211 MPa and 14 MPa, respectively), which has been extensively used as DE (Fig. R1).

Apart from the uniform network, long-chain crosslinker also tends to construct network with a considerable number of dissociative chains, which has been verified by gel test (Fig. 2b and Fig. R2). The influences of those dissociative chains on dielectric and mechanical properties of DE were particularly characterized by us for the first time. And among all the specimens we characterized in gel test, BAC2 contains the highest dissociative chains concentration (18.8 %) while VHBTM 4910 contains the lowest concentration (7 %). This is the reason why VHBTM 4910 and BAC2 were chosen to validate that dissociative chains in elastomer network can enhance dielectric constant while barely increase dielectric loss and mechanical loss (Fig. 2e, 2f and 2g).

The VHBTM 4910 is merely regarded as a most-frequently used reference for actuation performance and it was tested and compared throughout the work.

Fig. R1 a) Stress-strain curves of VHB™4910, BA-S, BA-M, BA-L and BAC2. b) Stress-strain curves of VHB™4910 and BAC series. c) Comparison of ultimate strength to modulus among VHB™ 4910, BA-S, BA-M, BA-L, BAC2, dielectric gel, multiblock copolymers, azo-grafted silicone, nitroaniline modified silicone, polyurethane, alkyl thiols grafted silicone and commercial Elastosil@Film. d) Toughness plotted against Young's modulus for VHB™ 4910, BA-S, BA-M, BA-L, BAC2, dielectric gel, silicones with ionic liquids, acrylic copolymer, nitroaniline modified silicone, nitrile modified silicone, alkyl thiols grafted silicone, 3D printable silicone, and commercial Elastosil@Film (*NPG Asia Mater.* 2018, 10, 821-826; *Chem. Eng. J.* 2021, 405, 126634; *ACS Appl. Mater. Interfaces* 2020, 12, 23432-23442; *ACS Appl. Mater. Interfaces* 2017, 9, 5237-5243; *J. Mater. Chem. C* 2018, 6, 2043-2053; *Adv. Eng. Mater.* 2019, 21, 1900481; *Polymer* 2018, 137, 269-275; *Macromol. Rapid Commun.* 2019, 40, 1900205; *Nat. Commun.* 2020, 11, 4000)

Fig. R2 Gel fraction and swelling ratio of VHB™4910, BA-S, BA-M, BA-L and BAC series.

2. In comparing with VHB - how did you decide how high a field to apply? What is the breakdown strength of your material, and how does it compare with VHB?

Our Response: Thanks for your critical questions. As for all actuation performance characterizations, electric field always rises from zero to the specific value which must be lower than breakdown strength at a given step. Furtherly considering the issue of insulation and safety, the maximum actuation field is set to 15 MV m^{-1} for actuation test without pre-strain and 70 MV m^{-1} with equiaxial pre-strain.

For breakdown strength, we are sorry that such a critical parameter was not measured and we have supplemented this measurement. A high voltage tester (BDJC-50kV, Beijing, Beiguang) with 25-mm-diameter pillar to plate electrode was used to characterize the electrical strength of elastomers. The elastomers were placed between two electrodes immersed in the silicone oil at room temperature. A DC voltage ramp of 500 V s^{-1} was applied to the electrodes until voltage drops sharply. Ten specimens were tested for VHB™4910 and BAC2 and the electrical breakdown strength (E_b) could be calculated using a two-parameter Weibull distribution function,

$$P = 1 - \exp(-(E_b/\alpha)^\beta)$$

where P is the cumulative probability of electric failure, E_b is the measured breakdown strength for each sample, α is the characteristic breakdown strength (characteristic E_b) that corresponds to a $\sim 63.2\%$ probability of failure, and β is the slope parameter that evaluates the scatter of data. Herein, the E_b was calculated from a linear fitting using Weibull failure statistics across 10 specimens per sample. These detailed electrical test conditions have also been added into the experimental section. Please refer to the **Revised Extended Data**.

Fig. R3 Comparison of electrical breakdown strength between VHB™4910 and BAC2 synthesized in this work.

As shown in Fig. R3, the electrical strength of BAC2 is 23.4 MV m^{-1} , a little lower than that of VHB™4910 (28.4 MV m^{-1}). Such a decrease in E_b could be attributed to the following aspects. Firstly, there is an empirical equation for evaluating electromechanical breakdown which derives from the deformation of soft elastomers caused by electrostatic compressive force, namely Maxwell Stress (*Prog. Mater. Sci. 2019, 100, 187-225*):

$$E_b = 0.6 \left(\frac{Y}{\varepsilon_0 \varepsilon_r} \right)^{\frac{1}{2}}$$

According to the equation, electrical strength of BAC2 could be estimated 44.5% lower than that of VHB™4910. In fact, the measured data demonstrates a much smaller difference (17.6%) in the breakdown strength of BAC2 and VHB™4910. Herein, electrical strength of BAC2 is satisfactory in view of its high dielectric constant and ultra-low modulus. Secondly, the geometry of the electrode of the high voltage tester has a great effect on the measurement of E_b . Generally, more test area of the sample during the measurement of E_b are used when employing the electrodes with a larger contact area, resulting covering more breakdown weak points in the sample and decreasing the measured value of E_b . Hence the E_b measured with the pillar to plate electrode is the lowest, compared with needle electrode and sphere electrode. The pillar to plate electrode is convenient to operate and therefore avoid mechanical damage for soft elastomers. Thirdly, It deserves noting that voltage drop, as the decisive condition for breakdown, is a little harsher than the current criterion (for example, $>5 \text{ mA}$), thereby reducing the measured value of breakdown strength.

3. It is stated that "As shown in Fig. 3d, the VHB™4910 elastomer needs almost 5 min to reach 90%

of its final strain". What is the reason for the low speed in both cases? Is it electrical connection, viscoelastic response or some other effect? How do you know? How much creep is there when the voltage is returned to zero? Can you bring the conclusions of your analysis in the supplementary material, to the main text?

Our Response: Thanks for your valuable comments. The measured response speed is susceptible to the test duration. For the low speed in static response property of BAC2 and VHBTM4910 (Fig.3d), viscoelastic response should bear main responsibility due to severe creep when compared with electrical connection. During the tests of actuation performance, an oscilloscope with a high voltage probe was employed to measure the voltage on elastomer films in real time. As shown in Fig. R4, film begins to expand when the voltage goes up (Fig. R4b) and continues expanding after the voltage reaches to the given value (Fig. R4c and R4d). In fact, area deformation was always detected in advance when compared to voltage augment. This abnormal phenomenon is ascribed to oscilloscope signal delay. Herein, the time voltage started to change was adjusted to the moment when active area began to vary, both in polarization process and in depolarization process. It deserves stating that for the sake of salient comparison, actuation deformation process under 70 MV m^{-1} rather than that under 40 MV m^{-1} is exhibited in Fig. R4 to demonstrate creep. After 600 s when voltage starts to descend, residual area strain is 49% and 74.7% for BAC2 and VHBTM4910 under 40 MV m^{-1} , respectively (Fig. R5).

Fig. R4 Actuation process of BAC2 from 0 to 70 MV m^{-1} . (Please note the voltage displayed on oscilloscope)

Fig. R5 Residual area strain for BAC2 and VHBTM4910 when electric field falls from 40 MV m⁻¹ to 0 (raw data).

In supplementary material, we gave some elaboration for actuation bandwidth from the view of current parameter. High voltage amplifier with limited output current, typically 1mA~2mA, will seriously constrain charging speed, resulting in low actuation bandwidth. This conclusion has been added to the main text, yet the detailed illustration and analysis has been reserved in Extended Data due to length limit. Please refer to **Revised Manuscript** and **Revised Extended Data**.

4. One of the challenges with VHB is its frequency response. While the new material presented here may be a little faster, what are the prospects for getting much higher frequency response?

Our Response: Thanks for your comments. Frequency response for BAC2 and VHBTM4910 to large drive signals (5 kV) in the range of 1 Hz to 100 Hz has been supplemented and shown in Fig. R6. Owe to suppressed mechanical loss, frequency response of BAC2 is nearly flat in the 1-100 Hz frequency range. Because the starting of resonance emerges at 100 Hz, resonance peak is reckoned to locate out of measurement range, which may need more attention in future. Rather, with the increase of frequency, actuation response of VHBTM4910 declines sharply, which, as a result, greatly limits it to applications below a few Hz.

Exactly, frequency response, which originates from severe viscoelasticity, is the greatest challenge for VHBTM4910 to applications. In my humble opinion, frequency response is mainly susceptible to mechanical loss, driving equipment and flexible electrodes. Firstly, reducing mechanical loss is principle and intrinsic for speed response, and there are three aspects that should be taken into consideration during elastomer synthesis: dynamic bonds like hydrogen bonds, dipole interaction and fillers. Dynamic bonds, for example hydrogen bonds, will repeatedly break and form during deformation, causing massive mechanical loss and slowing response speed (*J. Mater. Chem. C* 2019, 7, 12139-12150; *J. Mater. Chem. C* 2018, 6, 2043-2053;). Dipole interaction does the similar

work with hydrogen bonds (*J. Mater. Chem. C* 6, 2043-2053, 2018). Therefore, lots of elaboration and efforts should be put in selecting appropriate molecular structure. Fillers can destroy the integrity of crosslinking network, giving rise to sever viscoelastic loss (*Adv. Eng. Mater.* 21, 1900481, 2019). In a word, efforts should be made to decrease internal friction of chain segment movement. Moreover, preparation of acrylic elastomer with suppressed mechanical loss and fast response is our work in process. Secondly, polarization process is dependent on circuit parameters. And when charging time constant is comparable to viscoelastic relaxation time, reducing charging time constant can improve frequency response (*Adv. Funct. Mater.* 28, 1804328, 2018). Given this perspective, selecting voltage source with large power output and decreasing current-limiting resistance will do some help. Thirdly, the response speed of the actuator can also be influenced by the electrodes, which can contribute to the viscoelasticity of the device, and carbon grease demonstrates longest rise time when compared with ion implantation, conductive rubber (*Proc. SPIE* 8340 , 834004, 2012) and carbon black (*Adv. Funct. Mater.* 25, 1656–1665, 2015).

Fig. R6 Frequency response of samples to large drive signals (5 kV) in the range of 1 Hz to 100 Hz.

The data was normalized to 1 at 1 Hz.

5. More perspective would be valuable. For example, I don't see a discussion of work density. This is a very important property. Is it better than in VHB? I also don't see a comparison with actuation in other dielectric elastomer materials, which, for example, can offer much better frequency response. And there is little perspective given on the way forward. How do these materials fit within the theoretical performance criteria set out by Zhigang Suo? It is mentioned that frequency response can be improved - but some speculation on how much, and what this would mean for applications, would be helpful (added to the main text). Also - there is no discussion of breakdown or cycling/failure. Why not? What are the prospects here, compared to VHB and other materials?

Our Response: Thanks for your instructional suggestions. We have summarized and divided your

suggestions into four aspects: work density, frequency response, cycling test and model fitting, and they will be described and analyzed in detail based on supplementary tests and calculations.

Firstly, energy density was calculated based on measured data for actuation property (Fig. 3c) to demonstrate work density of our elastomer. Assuming that elastomers are incompressible and dielectric constant is invariable during deformation, estimated energy density ($1/2 e$) is obtained from equation below with constant-voltage drive (*Science 2000, 287, 836-839*):

$$(1 + s_z)^2(1 + s_A)^2 = 1$$

$$p = \varepsilon_0 \varepsilon_r E^2 = \frac{\varepsilon_0 \varepsilon_r E_0^2}{(1 + s_z)^2} = \frac{p_0}{(1 + s_z)^2} = p_0(1 + s_A)^2$$

$$e = -p(s_z + 0.5s_z^2)$$

where e , s_z , s_A , p and E , p_0 and E_0 are electromechanical energy density, thickness strain, area strain, Maxwell stress and electric field (divided by real-time thickness of film), initial Maxwell stress and electric field (divided by thickness of film before actuated), respectively.

As shown in Fig. R7, due to higher dielectric constant and strain, energy density of BAC2 is 242 kJ m⁻³, 5.78 times as much energy density as VHBTM4910 has under 70 MV m⁻¹, while mammalian skeletal muscles have an energy density of no more than 40 kJ m⁻³ (*IEEE J. Oceanic Eng. 2004, 29, 706-728*).

Secondly, to characterize the dynamic behavior, a frequency response analysis of BAC2 and VHBTM4910 with identical geometry and size, was performed. As shown in Fig. R6, with the increase of frequency, areal actuation strain of BAC2 decreases gently and then shows a slight raise while that of VHBTM4910 declines sharply, which can be attributed to lower mechanical loss. Herein, BAC2 could be employed in larger bandwidth cases than VHBTM4910 and speculated to have much higher power density in consideration of its ultra-high energy density. And later, cyclic test was performed on BAC2, with identical geometry and size to those adopted in frequency characterization. After 50 000 cycles, breakdown or failing did not emerge and actuation performance almost did not degrade (Fig. R8). Overall, BAC2 exhibits desirable dynamic response performance and has great potential for application in the range of at least 100 Hz. And for detailed information, please refer to **Revised Manuscript and Revised Extended Data**.

Fig. R7 Estimated energy density based on area strain with equiaxial pre-strain

Fig. R8 Cyclic actuation test of 50,000 cycles was performed on BAC2 film at electric field of 5 kV and excitation frequency of 5 Hz. The inset demonstrated z-axis displacement response in the last five cycles.

Professor Suo in Harvard University have set out the theoretical performance criteria which predicts that an elastomer capable of giant deformation should have a stress-stretch curve of the following desirable features: The elastomer is soft at small stretch ratios and stiffens sharply at moderate stretch ratios (*Phys. Rev. Lett.* 2010, 104, 178302). Therefore, Gent model (*Rubber Chem. Techno.* 1996, 69, 59-61) was used to obtain stress-stretch curve of elastomer films under biaxial stresses from measured stress-stretch curve under uniaxial stresses. An empirical constitutive relation, two-constant, was proposed by Gent as following:

$$W = -\frac{E}{6} J_m \ln\left(1 - \frac{J_1}{J_m}\right)$$

where W is elastically-stored strain energy density, E denotes the small-strain tensile modulus, and J_1 is the first strain invariant, defined as:

$$J_1 = \lambda_1^2 + \lambda_2^2 + \lambda_3^2 - 3$$

where λ_1 , λ_2 and λ_3 are the principal stretch ratios. And J_m is a maximum value of J_1 , at which the film reaches a limiting state.

For a uniaxial extension, $\lambda_1 = \lambda$, $\lambda_2 = \lambda_3 = \lambda^{-1/2}$, $J_1 = \lambda^2 + 2\lambda^{-1} - 3$ and $t_2 = t_3 = 0$. Thus, true stress in extension direction is

$$t_1 = \frac{E (\lambda^2 - \lambda^{-1})}{3 (1 - J_1/J_m)}$$

And nominal stress ($\sigma(\lambda)$) is given by t_1/λ .

Based on the least-square method, E and J_m are estimated to be 0.06914 MPa and 360 for VHBTM4910, and 0.03503 MPa and 700 for BAC2, respectively. Fitted curves and measured curves were demonstrated in Fig. R9

Fig. R9 Uniaxial stress-stretch measured curves and fitted curves based on Gent model.

For biaxial extension, $\lambda_1 = \lambda_2 = \lambda$, $\lambda_3 = \lambda^{-2}$, $J_1 = 2\lambda^2 + \lambda^{-4} - 3$ and $t_3 = 0$. Thus, true stress in extension direction is

$$t_1 = t_2 = \lambda_1 \frac{\partial W}{\partial \lambda_1} - \lambda_3 \frac{\partial W}{\partial \lambda_3} = \frac{E (\lambda^2 - \lambda^{-4})}{3 (1 - J_1/J_m)}$$

Then, parameters fitted in Fig. R8 were substituted into equation above to compute stress-stretch curves under biaxial stresses, and the results are drawn in Fig. R10.

Fig. R10 Calculated stress-stretch curves of VHBTM4910 and BAC2 under biaxial stresses based on Gent model.

Finally, according to the equation set proposed by Suo et al (*Phys. Rev. Lett.* 2010, 104, 178302):

$$\Phi(\lambda) = H\lambda^{-2}\sqrt{t_1(\lambda)/\varepsilon_0\varepsilon_r}$$

$$\Phi_B(\lambda) = E_B H\lambda^{-2}$$

voltage-stretch curves are plotted in Fig. R11 and R12 using calculated data above and experimental values $H=1$ mm, $E_B=28.4$ MV m⁻¹ and $\varepsilon_r=4.43$ for VHBTM4910, and $E_B=23.4$ MV m⁻¹ and $\varepsilon_r=5.75$ for BAC2, respectively.

Fig. R11 Voltage-stretch curve of VHBTM4910 plotted according to Suo 's theory.

Fig. R12 Voltage-stretch curve of BAC2 plotted according to Suo's theory.

As drawn in figures, voltage-stretch curves indicate that BAC2 is a type II dielectric while VHBTM4910 is a type I dielectric. We regret that BAC2 does not conform with the theoretical performance criteria proposed by Zhigang Suo due to its ultra-high stretchability and slow stiffness in mediate stage of stretch. Actually, with the increase of stretch, E_B of VHBTM4910 and BAC2 rises up, of which the mechanism is not clear now. Additionally, although Gent model gives a decent fit to elastic behavior of materials over the entire range of possible extension, extension information is generally not enough to estimate biaxial stress data accurately. Thus, the discussion stated here is remarkably simple. And pre-strain may be necessary to enhance electromechanical stability and attain giant deformation (*App. Phys. Lett.* 2007, 91, 061921).

6. Overall I believe this work is making a significant contribution, but it is incomplete, and the authors' reasonable hypotheses on the underlying mechanisms are not well justified in the main text.

Our Response: Thanks for your valuable suggestions. The hypothesis that employing macromolecular crosslinkers whose average relative molecular weight matches crosslink density of polymer network can highly improve its uniformity has been verified through comparison of elastomers crosslinked by rationally-selected crosslinkers with different molecular weight. Firstly, CN9021NS ($\overline{M}_n=28000 \text{ g mol}^{-1}$), a difunctional urethane acrylate compound composed of a flexible polyether diol segment and an aliphatic diisocyanate segment, was chosen as macromolecular crosslinker for the construction of *n*-butyl acrylic-based elastomer network (BAC series). For comparison, *n*-butyl acrylate homopolymer (BA-S, BA-M and BA-L), crosslinked by the equimolar (taking BAC2 as reference) polyethylene glycol diacrylate (a small molecular crosslinker, $\overline{M}_n=575 \text{ g mol}^{-1}$), CN9893NS (a medium-molecular difunctional urethane acrylate crosslinker, $\overline{M}_n=1600 \text{ g mol}^{-1}$) and CN9014NS (a large-molecular difunctional urethane acrylate crosslinker, $\overline{M}_n=6800 \text{ g mol}^{-1}$), respectively, was also prepared through a similar polymerization process. Considering that chemical reactivity of crosslinkers is very susceptible to their neighboring groups, BA-S, BA-M, and BA-L, whose crosslinker has the same groups bonding to crosslinking points as CN9021NS, have been synthesized. And BAP in Original Manuscript has been renamed as BA-S in Revised Manuscript. Secondly, four formulations with different contents of CN9021NS as the crosslinker have been systematically characterized and demonstrated in **Revised Extended Data**, among which BAC2 exhibits an optimum comprehensive performance including the highest actuation sensitivity and superior ductility. As shown in Fig.1c to Fig.1e, BAC2 further precedes BA-S, BA-M and BA-L in stretchability due to a more uniform crosslinking network, which was achieved through rationally selecting macro-molecular crosslinker. Generally, polymer with low elastic modulus (especially lower than 0.1 MPa, like gels) tends to show poor ductility and terrible ultimate strength, while BAC2 synthesized in this work have all those satisfied properties (Fig.1d and 1e). Moreover, after measuring the true cross-sectional area of specimen, we correct the ultimate strength of BAC2 from 1.34 MPa (nominal stress) to 32 MPa. Such a value, to the best of our knowledge, is the highest value among those soft polymers (*Chem. Eng. J.* 2021, 405, 126634). The BAC2 is 60% lower in Young's modulus (0.073 MPa) but 128% higher in tensile strength (32 MPa) than that of VHBTM4910 (0.211 MPa and 14 MPa, respectively), which has been extensively used as DE.

And in order to verify the influence of those dissociative chains, dielectric and mechanical properties

have been characterized and compared between elastomers before and after swollen to remove uncrosslinked chains. The results demonstrates that those dissociative chains can increase dielectric constant without sacrificing low loss. At your suggestion, lots of modifications and supplementary experiments have been done and then our understanding and thinking goes deeper. Please refer to **Revised Manuscript** and **Revised Extended Data**. And these hypotheses will be followed and further investigated in later work.

Reviewer #2: The manuscript reported an optimizing crosslinking strategy to prepare a dielectric elastomer with low loss and outstanding actuation performance. The strategy is clearly presented. The dielectric, mechanical and actuation performance of the elastomers are demonstrated by experiments. The results are solid and interesting, but clear major flaws exist and I cannot recommend its publication in the current form. The following questions should be carefully addressed.

Our Response: Thanks for your reasonable comments and valuable suggestions. We have supplemented some experiments according to your constructive recommendations and then made necessary modifications carefully. First of all, considering that chemical reactivity of crosslinkers is very susceptible to their neighboring groups, BA-S, BA-M, and BA-L, whose crosslinker has the same groups bonding to crosslinking points as CN9021NS, have been synthesized. And BAP in Original Manuscript has been renamed as BA-S in Revised Manuscript. Please refer to the replies below and the **Revised Manuscript** as well as **the Revised Extended Data**.

1. The authors claimed the newly synthesized material has low loss, as seen in the title of the paper. However, as shown in Fig. 2h, the hysteresis is still large. The characterization of mechanical loss is too limited and no mechanism was discussed.

Our Response: Thanks for your comments. Low loss, which is a desirable feature for the new synthesized material and brings it great practical value, has two implications, including low dielectric loss and low mechanical loss. Specifically, dielectric loss factor (Fig. 2e) among the most commonly-used frequency range of 100 Hz to 1500 Hz is below 0.2 %, which, to the best of our knowledge, is lowest among those elastomers with a relative dielectric constant higher than 5 (*Macromol. Rapid Commun.* 2018, 39, 1800340; *Macromol. Rapid Commun.* 2019, 40, 1900205; *ACS Appl. Mater. Interfaces* 2020, 12, 23432-23442; *J. Mater. Chem. C*, 2018, 6, 2043-2053; *Adv. Eng. Mater.* 2019, 21, 1900481).

Hysteresis calculated by stress-strain loops in Fig. 2h is badly dependent on stretch rate. Mechanical loss factor ($\tan \delta$) measured by DMA is more comparable with previous materials. BAC2 ($\tan \delta \sim 0.21$

@ 20°C, 1 Hz) presents much advantages over VHB™ 4910 (tan δ ~0.93 @ 20°C, 1 Hz) and other acrylic elastomers (tan δ ~0.4 @ 20°C, 1 Hz for copolymer of acrylate and polyurethane in *Polymer* 2018, 149, 39-44; tan δ ~ 0.4~1.0 @ 20°C, 1 Hz for acrylate in *Polymer* 2018, 137, 269-275). Generally, because glass transition temperature approaches room temperature, mechanical loss of acrylic elastomers is much higher than that of silicones. Thus, mechanical loss factor of BAC2 is also close to that (tan δ ~0.209 @ 20°C, 1 Hz) of Sylgard 184 with similar elastic modulus, a widely-used commercial silicone from Dow Corning (*Adv. Funct. Mater.* 2018, 28, 1804328). Hence, our declaration that our newly synthesized elastomer has low loss is well-grounded.

In order to reduce mechanical loss, there are three aspects that should be taken into consideration during elastomer synthesis: dynamic bonds like hydrogen bonds, dipole interaction and fillers. To be specific, dynamic bonds, for example hydrogen bonds, will repeatedly break and form during deformation, causing massive mechanical loss and slowing response speed (*J. Mater. Chem. C* 2019, 7, 12139-12150; *J. Mater. Chem. C* 2018, 6, 2043-2053). Dipole interaction does the similar work with hydrogen bonds (*J. Mater. Chem. C* 2018, 6, 2043-2053). Therefore, lots of elaboration and efforts should be put in selecting appropriate molecular structure. Fillers can destroy the integrity of crosslinking network, giving rise to sever viscoelastic loss (*Adv. Eng. Mater.* 2019, 21, 1900481). In a word, efforts should be made to decrease internal friction of chain segment movement. Therefore, the desirable feature for BAC2 is attributed to the absence of hydrogen bonds and strong polar dipoles as well as fillers. Besides, flexible long aliphatic polyether backbone inside CN9021NS also acts as “lubricant” to decrease interaction between chain segments.

2. The authors claimed the newly synthesized material has outstanding actuation performance. This conclusion was drawn based on the experimental results in Fig. 3b, 3c, where the actuation performance was compared with the commonly used DE material VHB. However, the actuation strain of VHB achieved by the authors was too small compared to those in literature, and it was apparently not a fair comparison. To the reviewer’s opinion, the actuation performance of the new material is not as good as VHB. See a recent review about the actuation strain of VHB. “Mechanics of dielectric elastomer structures: A review, Extreme Mechanics Letters 2020, 38, 100752”.

Our Response: Thanks for your valuable suggestion and recommendation of an important latest review in this field, which has been studied seriously. And after that, two reasons why actuation strain of VHB in literature was larger than that in our work are summarized: much higher actuation electric field and elaborate techniques or ingenious structure. For example, Pelrine et al used VHB 4910 to

obtain 158% circular strain under 412 MV/m and 215% linear strain under 239 MV/m with prestrain, respectively (*Science* 2000, 287, 836-839). Huang et al has achieved 488% areal strain under 5.25 kV (the value was drawn from figure) with dead load. The equal-biaxial dead load has been proved to suppress electric breakdown and consequently enables elastomers to gain a greater voltage-induced deformation than rigid constraint does. (*Appl. Phys. Lett* 2012, 100, 041911). Xu et al has demonstrated 230% area strain based on sulfonated pentablock copolymer with a driving voltage of 3.5 kV. Additionally, the devices possessed a relatively small active area with a wrinkled surface before electrical actuation, but a larger active area with a flattened surface after electrical actuation (*Science* 2018, 359, 1495–1500).

In our work, the maximum actuation electric field is merely 70 MV/m and testing apparatus has been simplified to display elastomer performance only. In fact, approximately 30% area strain @70 MV/m for VHB™ 4910 obtained here is not too small compared with those in literature under the same electric field (*Appl. Phys. Lett* 2012, 100, 041911; *Soft Matter* 2012, 8, 6167-6173). Additionally, the commonly-used maximum area actuation strain for VHB™4910 is 7.5% @ 17 MV/m without pre-strain, which was reported by Spontak (*Macromol. Rapid Commun.* 2007, 28, 1142–1147; *Polymer* 2018, 137, 269-275; *Chem. Eng. J.* 2020, 382, 123037). Then, taking this value as reference, area strain (18.5% @ 15 MV/m) for BAC2 still has its advantages.

Therefore, higher dielectric constant and much lower modulus enable our new elastomer to obtain larger actuation strain-118% @70 MV/m and 18.5% @15 MV/m, demonstrating far better performance than VHB.

3. In Fig.3(b-d) and Extended Data Fig.5(b-f), the authors claimed that the optimizing network helps to obtain excellent dielectric and actuation performance. However, in these figures, the performances of BAP are not presented to show how the optimizing crosslinking strategy can improve the dielectric and actuation performance.

Our Response: Thanks for your comments. As we have said at the beginning, And BAP in Original Manuscript has been renamed as BA-S in Revised Manuscript. Therefore, we are going to demonstrate the performances of BA-S, BA-M and BA-L.

Fig. R1 Stress-strain curves of BA-S, BA-M, BA-L and BAC2 until fracture a) and 100% strain b). Tangent lines at 5% strain were drawn in figure showing elastic moduli. c) Frequency dependence of dielectric properties and actuation sensitivity d) for BA-S, BA-M, BA-L and BAC2.

According to the Maxwell stress, the actuation strain (S_z) (*Science* 2000, 287, 836-839) in the thickness direction can be defined by $S_z = -\frac{\epsilon_0 \epsilon_r E^2}{Y}$, where ϵ_0 , ϵ_r , Y and E are the dielectric constant of vacuum, dielectric constant of elastomer, Young's modulus of elastomer and applied electric field, respectively. As shown in Fig. R1, dielectric constant shows slight difference among BA-S, BA-M, BA-L and BAC2, while elastic modulus at 5% strain for BAC2, which was the slope of the tangent line at 5% strain from stress-strain curve, is much lower than those of the others. Accordingly, BAC2 presents the highest actuation sensitivity (defined as $\beta = \frac{\epsilon_r}{Y}$). Therefore, we can make an informed inference that actuation strain of BA-S, BA-M and BA-L is inferior to that of BAC2 without pre-strain. Additionally, we can find that the elastic modulus for BA-S, BA-M and BA-L is further higher than that of BAC2 at a large stretch (Fig. R1a)), so our inference is still well-founded for actuation with pre-strain. In fact, BA-S, BA-M and BA-L is not able to bear 400% equiaxial pre-strain. In conclusion, BAC2 presents excellent dielectric and actuation performance compared to BA-S, BA-

M, BA-L as well as VHBTM4910, and it can be reasonably attributed to its macro-molecular crosslinker whose average molecular weight well matches the range of average molecular weight (\overline{M}_c) between crosslinking points.

4. In Extended Data Table 1, weight ratio of CN9254NS is over 30%, most of which are not reacted. Do the unreacted CN9254NS leak like the plasticizers? Given that there is a large amount of double bonds in the unreacted CN9254NS, do these double bonds make the dielectric elastomer unstable in actuation performance after long-time service?

Our Response: Thanks for your comments. Firstly, we would like to make some clarifications that most of macro-molecular crosslinker-CN9021NS (not CN9254NS) are reacted. This conclusion is strongly supported by the results of gel test and elemental analysis that the weight ratio of uncrosslinked chains is about 15%~19% (Fig.2b), approximately half of which is from CN9021NS (Fig.2c).

As for your enlightening concerns about leakage of those uncrosslinked chains and long-term actuation stability, we have investigated thoroughly from three aspects. Firstly, cyclic actuation test of 50,000 cycles was performed without performance deterioration. Z-axial displacement has undergone a mild growth firstly, which may be ascribed to its own gravity, and then fallen back slowly to the level at first several cycles and remain stable (Fig. R2).

Fig. R2 Cyclic actuation test of 50,000 cycles was performed on BAC2 film at electric field of 5 kV and excitation frequency of 5 Hz. The inset demonstrated z-axis displacement response in the last five cycles.

Secondly, in order to further verify whether those uncrosslinked chains leak like plasticizer, a specially-designed gel test was performed for BAC2 newly synthesized and after storage for 125 days. Samples were stirred in tetrahydrofuran (THF) for 1 min to remove uncrosslinked chains at the

surface layer, then drying in oven at 40 °C for 24h. The residual mass of newly-synthesized BAC2 is 98.55% while the residual mass of BAC2 after storage for 125 days is 98.64%. Hence, those uncrosslinked chains do not leak like plasticizer.

Thirdly, to study long-term stability, a whole set of dielectric and mechanical, even as well as actuation testing was performed on BAC2 after storage at atmosphere environment for 125 days. Results are summarized and exhibited in Fig. R3. After storage for 125 days, elastic modulus of BAC2 rises slightly while elongation and ultimate strength drop off a little which also appears for BA-S, BA-M and BA-L (Fig. R3a). Accordingly, actuation sensitivity of BAC2 after storage for 125 days declines somewhat due to almost no difference in dielectric constant. Additionally, actuation area strain comparison between BAC2 and BAC2 after storage 125 days confirms this result again. Frankly, we should admit that mechanical property of BAC2 has deteriorated to a certain extend after storage at atmosphere environment for 125 days, resulting in a little decline in actuation performance. Yet, the change is prevailing for acrylic elastomers without further treatments and mechanical properties as well as actuation performance of BAC2 after storage for 125 days are still far superior to BA-S, BA-M and BA-L (Fig. R1 and Fig. R3). Besides, these results cannot be attributed to a small amount of unreacted CN9021NS, and we will further study in the future.

Fig. R3 a) Comparison of stress-strain curves for BA-S, BA-M, BA-L and BAC2 newly synthesized and those after storage for days. b) Comparison of dielectric properties for newly-synthesized BAC2 and BAC2 after storage for 125 days. c) Actuation sensitivity of VHBTM 4910, BAC2 and BAC2 after storage for 125 days. d) Actuation area strain of newly-synthesized BAC2 and BAC2 after storage for 125 days.

In a word, those uncrosslinked chains will not leak like plasticizer in cyclic actuation and long-time service, while actuation performance of BAC2 has a little deteriorated after storage at atmosphere environment for 125 days. The cause is yet ambiguous.

5. The mechanical tests in the paper seemed not be professional. In the supporting information (Line 86), why 500 is chosen to be the stretch rate for Young's modulus test? Why are the stretch rate for Young's modulus test, fracture test and stress-strain loops not the same (Line86-91 supporting information)? The material should be rate-dependent.

Our Response: Thanks for your careful suggestions. Young's modulus was characterized first, and sample was cut into large-size dumbbell-type 1A to reduce the measurement error greatly. Then according to ISO 37, the stretch rate of 500 mm min⁻¹ is recommended. However, for fracture test, the displacement range of universal mechanical machine-Instron 3343 is too small to snap the type 1A dumbbell shape sample. Besides, to reduce the effect of the dumbbell shoulder, the fracture test was performed on a dumbbell shape with a smaller size at the rate of 200 mm min⁻¹ with reference to the published work of Pei et al (*J. Polym. Sci., Part B: Polym. Phys.* 2013, 51, 197-206). For stress-strain loop test, as the stretch rate is relatively low, dumbbell-shape samples were replaced by strip ones to eliminate the effect of the dumbbell shoulder. And both the sample scale and stretch rate referred to previous work of our group (*Polymer* 2018, 137, 269-275). In order to compare our new elastomers with advanced materials precisely, the experiment setup referred to international standards and advanced studies as far as possible. As what you have stated, the elastomers are rate-dependent. That is also the reason why we set different experiment parameter and the consideration may be not thoughtful at the same time. We have made corrections to elastic moduli and redefined them as the slopes of tangent lines at 5% strain on stress-strain curves. Thanks for your scrupulous review and comment. We will consider mechanical test more tender in future research.

6. In the manuscript, the authors claimed that the BAC2 sample has a high toughness of 6.77J/m³, and Fig. 1e shows the diagram of toughness. However, in mechanics test, toughness has a standard definition by measuring a sample with a pre-cut crack and has a unit of J/m².

Our Response: Thanks for your comments. To the best of our knowledge, fracture energy (Γ) which is measured by stretching a sample with a pre-cut crack and has a unit of J/m^2 , just as you have stated, is usually used to describe tear resistance and fatigue resistance of soft materials, especially for gels (*Adv. Mater.* 2018, 30, 1706846; *Nat. Commun.* 2020, 11). Suo et al have proposed a method to determine the fracture energy of stretchable gels (*Nature* 2012, 489, 133-136). However, the experiments showed that the measured fracture energy is dependent of the shape and size of the specimens (*Nature* 2012, 489, 133-136; *J. Appl. Phys.* 2012, 111, 104114). Toughness we used in this work is more forced on rupture property which is more relevant to practical applications. In some degree, the influence of the shape and size of the specimens on toughness is eliminated during calculation. In fact, both fracture energy and toughness can be employed to characterize the mechanical strength of stretchable materials. Sometimes, they appear simultaneously in a paper, and change with the same trend (*J. Mater. Chem. B* 2018, 6, 8105-8114; *Adv. Mater.* 2018, 30, 1705145; *Polymer* 2010, 51, 4152-4159; *ACS Appl. Mater. Interfaces* 2018, 10, 26610-26617; *Nat. Commun.* 2020, 11, 4000). In future research, we will pay more attention to fracture energy. Thanks for your comprehensive comments and suggestions again.

Reviewer #3: This paper reports the design of polyacrylate elastomers for dielectric actuation. The prepared materials are slightly softer, more extendable, and less viscous than commercial VHB elastomers. Therefore, their actuation properties are slightly better than VHB, which cannot be viewed as revolutionary. Also, the idea behind the pursued materials design strategy needs better justification. Currently, it looks like a trial-and-error approach. What is the major innovation in this paper?

Our Response: Thanks for your comments. In this contribution, we report a new polyacrylate dielectric elastomer with optimized polymer network by rationally employing difunctional macromolecular crosslinking agent via a simple UV curing synthesis process. The synthesized acrylic elastomers exhibit relatively high dielectric constant, ultra-low dielectric loss and mechanical loss, and desirable Young's modulus, which are superior to commercial 3M VHBTM 4910. Based on improved dielectric and mechanical performance, large actuation strain and rapid response are successfully achieved in this elaborate polyacrylate elastomer. Besides, a home-made motor made of synthesized polyacrylate dielectric elastomer could be driven under a low electric field of 32 MV m^{-1} and high frequency. Especially, the rotation speed is 15 times higher than that of 3M VHBTM 4910-based motor. Specific test data has been summarized in detail in Table R1. Besides, it is a huge

challenge to simultaneously improve the softness and toughness of elastomer and such a paradox is solved in this contribution through the optimization of crosslinked network. The BAC2 sample not only exhibits a low Young's modulus (0.073 MPa) and a high elongation (2400%) but also displays a high toughness (6.77 MJ m⁻³) and a high strength (32.23 MPa). The excellent mechanical property displayed here is seldom observed in previously reported works. Conventionally, most strategies with an effort of contributing to one performance improvement (i.e., low modulus, high dielectric constant, low mechanical loss) for acrylic elastomers, may cause the degradation of the others (*J. Mater. Chem. C* 2019, 7, 12139-12150; *J. Mater. Chem. C* 2018, 6, 2043-2053; *Adv. Eng. Mater.* 2019, 21, 1900481). The design and preparation of high-quality dielectric elastomers remains a challenging problem for the mass application of DEAs. Here, we innovatively select macro-molecular crosslinker to construct network to realize significant improvement of the whole performance, which exactly is revolutionary. Based on the above-mentioned properties, we prudently raise some objections to your harsh criticism that actuation properties of BAC2 are slightly better than VHB, which cannot be viewed as revolutionary.

Table R1 Comparison of performances between VHBTM4910 and BAC2

	VHB TM 4910	BAC2
Dielectric constant @ 1kHz	4.4	5.75
Dielectric loss @ 1kHz	0.0229	0.002
Young's modulus at 5% strain (MPa)	0.211	0.073
Mechanical loss factor (tan δ) @ 20°C, 1 Hz	0.21	0.93
Actuation sensitivity (MPa ⁻¹)	20.995	78.8
Elongation at break (%)	~1500	~2400
Ultimate true strength (MPa)	14.2	32.23
Ultimate true strength to modulus	67.32	441.53
Toughness (MJ m ⁻³)	4.37	6.77
Area strain @ 15 kV m ⁻¹ without pre-strain	4.5%	18.5%
Area strain @ 70 kV m ⁻¹ with pre-strain	34%	118%
Estimated energy density (MJ m ⁻³)	0.042	0.242
Frequency bandwidth (Hz)	<10	>100
Motor speed (r s ⁻¹)	0.045	0.72

In addition, in order to further verify the influence of molecular weight of crosslinkers, *n*-butyl acrylate homopolymer (BA-S, BA-M and BA-L), crosslinked by the equimolar (taking BAC2 as reference) polyethylene glycol diacrylate (a small molecular crosslinker, $\overline{M}_n=575 \text{ g mol}^{-1}$), CN9893NS (a medium molecular crosslinker, $\overline{M}_n=1600 \text{ g mol}^{-1}$) and CN9014NS (a large molecular crosslinker, $\overline{M}_n=6800 \text{ g mol}^{-1}$), respectively, was also prepared and characterized. The huge difference between BAC2 and BA-S, BA-M and BA-L on mechanical properties confirms again that selecting crosslinker whose molecular weight matches the average molecular weight among crosslinking points (\overline{M}_c) of elastomers can improve their performance. The approach mentioned here is crucially important for the development of high-performance dielectric elastomers with remarkable actuation capability.

Besides, some major modifications need explanation in advance. Considering that chemical reactivity of crosslinkers is very susceptible to their neighboring groups, BA-S, BA-M, and BA-L, whose crosslinker has the same groups bonding to crosslinking points as CN9021NS, have been synthesized. And BAP in Original Manuscript has been renamed as BA-S in Revised Manuscript. Please refer to the replies below and the **Revised Manuscript** as well as **the Revised Extended Data**.

1. Line 15: Why is 0.088 MPa considered as a desirable Young's modulus? Would a modulus of 0.01 or even 0.001 MPa be more desirable? I guess, it depends on application.

Our Response: Thanks for your kind comments. According to another reviewer's suggestion, Young's modulus has been redefined as the slope of tangent line at 5% strain on stress-strain curves and therefore correct from 0.088 MPa to 0.073 MPa. Now, we would like to give some explanations about 0.073 MPa.

According to the Maxwell stress, the actuation strain (S_z) (*Science 2000, 287, 836-839*) in the thickness direction can be defined by $S_z = -\frac{\varepsilon_0 \varepsilon_r E^2}{Y}$, where ε_0 , ε_r , Y and E are the dielectric constant of vacuum, dielectric constant of elastomer, Young's modulus of elastomer and applied electric field, respectively. Thus, the driven E under a fixed actuation strain can be reduced by either increasing the ε_r or decreasing the Y of the elastomer. Hence, reduction of modulus from 0.211 MPa to 0.073 MPa can enhance actuation strain and lower electric driving field (Fig. 3b, 3c and Fig. 4). Nevertheless, it does not mean that the softer the elastomer is, the better the performance is. Too soft materials will

bring about troubles in three aspects: mechanical strength, electrical breakdown, and output force. Firstly, soft elastomer usually is delicate and fragile, and therefore easy to rupture, which presents difficulty for its applications (*Adv. Funct. Mater.* 2015, 25, 1656–1665). Secondly, there is an empirical equation for evaluating electromechanical breakdown which derives from the deformation of soft elastomers caused by electrostatic compressive force, namely Maxwell Stress (*Prog. Mater. Sci.* 2019, 100, 187-225):

$$E_b = 0.6 \left(\frac{Y}{\varepsilon_0 \varepsilon_r} \right)^{\frac{1}{2}}$$

Softness will deteriorate insulating property badly. Last, the softer the elastomer is, the smaller the output force is. Too small force will not drive devices. Therefore, elastic modulus of elastomer has an optimal value which is dependent on the application, actuation parameters and working conditions. Modeling and discussion on this question is not the focus of this contribution, and will be noted in the following studies.

In this work, Young's modulus has achieved lowest value to 0.073 MPa by tuning crosslinker content, and meanwhile, ultimate true strength and toughness are higher than others (Fig. R1). Besides, mainly due to much lower modulus, BAC2 presents larger actuation strain and higher electro-mechanical energy density (242 kJ m⁻³, 5.78 times as much energy density as VHBTM4910), and hence, the motor fabricated by it need lower driving voltage and outputs more mechanical power. As a result, the Young's modulus of BAC2 (0.073 MPa) is considered desirable.

Fig. R1 a), Stress-strain curves of VHBTM4910, BAC series samples. b), Comparison of actuation sensitivity, dielectric constant and Young's modulus of VHBTM4910 and BAC series samples.

2. In line 16, 118% should not be considered as a “huge actuation strain”. Many DEA systems show larger strains at lower fields (70 MV/um is a relatively high field).

Our Response: Thanks for your comment. The actuation field is 70 MV m^{-1} rather than 70 MV um^{-1} , which needs some clarification. Due to improved actuation properties, BAC2 elastomer exhibits much larger area strain of 18.5% at 15 MV m^{-1} (nominal electric field) without pre-strain and 118% area strain at 70 MV m^{-1} (nominal electric field) with equiaxial pre-strain while the area strain obtained from VHB™ 4910 film at the same electric field is only 4.5% and 34%, respectively (Fig. 3b, 3c). It may be somewhat inappropriate that we considered 118% at 70 MV m^{-1} as huge actuation strain, but BAC2 in actuation performance is greatly superior to VHB™ 4910 and many previous elastomers, as summarized in Table R2. Strictly, the expression has been changed to large actuation strain in **Revised Manuscript**.

Table R2 Comparison of actuation performance of BAC2 with that of advanced DE materials reported in the literature

sample	Dielectric constant @1 kHz	Young's modulus (MPa)	Actuation sensitivity (MPa^{-1})	Area strain	Pre-strain (%)	reference
Azo-grafted PDMS	<4.9	0.725	<6.75	<18%@70MV/m	-	J. Mater. Chem. C 2015, 3, 4883-4889
Acrylic copolymer	~7	~0.17	41	~14%@15MV/m	(0,0)	Polymer 2018, 149, 39-44
CN9021-PEGDA	9.4	0.323	29	17%@15MV/m	(0,0)	NPG Asia Mater. 2019, 11, 62
SBAS triblock copolymer	4.8	0.39	12.3	<10%@15MV/m 72%@154MV/m	-	Chem. Eng. J. 2020, 382, 123037
LSR4305	2.9	-	-	10%@15MV/m 25%@25MV/m	-	Adv. Funct. Mater. 2020, 2006639
Polysiloxanes with polar groups	~5.5	0.41	13.4	5%@30MV/m	-	J. Mater. Chem. C 2019, 7, 12139.
(SEHAS)2	3.8	0.11	34.5	<10%@15MV/m 150%@75MV/m	(0,0) (150,150)	Chem. Eng. J. 405, 126634, (2021)
nitroaniline modified silicone	10.6	0.27	40	6.5%@7MV/m	(7.5,7.5)	ACS Appl. Mater. Interfaces 2020, 12, 23432–23442
Elastosil® Film	2.9	1.2	2.4	16.6%@90MV/m	(7.5,7.5)	J. Mater. Chem. C 2018, 6, 2043--2053
VHB™4910	4.2	0.35	12	7.5%@17MV/m	(0,0)	Macromol. Rapid Commun. 2007, 28, 1142–1147
VHB™4910 in this work	4.4	0.211	~21	4.5%@15MV/m ~30%@70MV/m	(0,0) (400,400)	
BAC2 in this work	5.75	0.073	78.8	18.5%@15MV/m 118%@70MV/m	(0,0) (400,400)	

3. The modulus of 88 kPa corresponds to the lower limit for conventional polymer networks controlled by chain entanglements. Furthermore, enhancement of chain flexibility (by e.g., introducing polyether diol segments in CN9021NS crosslinker) promotes entanglements and therefore raise the lower limit for network modulus. In other words, the design strategy discussed in lines 75-82 is not “helpful to

achieving low elastic modulus”.

Our Response: Thanks for your comments. Here, we would like to make an explanation on this issue from three aspects.

Firstly, Although, reducing Young’s modulus can improve actuation property, low-density crosslinking is necessary to ensure high elasticity. If there are only physical entanglements among poly(*n*-butyl acrylate) chains, the slippage appears under large deformation. Hence, the theoretical minimum modulus based on entanglements is unpractical.

Secondly, the modulus measurements depend on sample size and shape, tensile rate, strain, and so on. Besides, Much lower moduli, like 25 kPa for commercial silicones (*Adv. Funct. Mater.* **2018**, **28**, **1804328**), 48kPa for PU/CNT composite (*Mater. Sci. Eng., C* **2007**, **27**, **110**) as well as 40 kPa for bistable electroactive elastomer (*ACS Appl. Polym. Mater.* **2020**, **2**, **2008–2015**), have been reported. Therefore, we think it is a pure coincidence that the Young’s modulus of BAC2 (0.088 MPa) is exactly equal to the theoretical minimum modulus (88 kPa) based on entanglements, and it is maybe a little inappropriate for modulus to strictly compare experimental data with theoretical values.

Thirdly, in this contribution, CN9021NS helps to achieve low elastic modulus due to reduced crosslinking density and high elongation due to suitable molecular weight. Therefore, low elastic modulus is attributed to the high flexibility of poly(*n*-butyl acrylate) chains and polyether diol segments in CN9021NS, and reduced degree of crosslinking. More distinct exposition has been supplemented to substitute the original expression. Please refer to the **Revised Manuscript**.

Material ^{a)}	Modulus ^{b)} [kPa]	$\tan \delta$ @ 1Hz ^{c)}	$\tan \delta$ @ 10Hz ^{c)}	$\tan \delta$ @ 100Hz ^{c)}
Ecoflex 0030	105.9	0.085	0.140	0.169
E-S mixture 2:1 ^{d)}	99.5	0.086	0.179	0.270
E-S mixture 1:1 ^{d)}	77.6	0.105	0.235	0.343
E-S mixture 1:2 ^{d)}	70.1	0.112	0.264	0.414
Sylgard (40:1)	82.7	0.206	0.402	0.612
Sylgard (50:1)	25.4	0.298	0.605	0.903
VHB ^{e)}	299.9	0.659	0.997	1.176

^{a)}All samples were cylinders with 1 cm height and 1 cm diameter and were fully cured at temperature of 70 °C for 1 h; ^{b)}modulus was tested in compression using an Instron; ^{c)} $\tan \delta$ were tested in compression with a 100 mN precompression and dynamic mechanical data were collected under 1% strain, using a Bose 3200 DMA system; ^{d)}E: Ecoflex 0030, S: Sylgard 184 (40:1), the mixtures were prepared using a Thinky mixer; ^{e)}VHB samples were prepared by rolling VHB 4905 film into cylinder of 1 cm diameter and 1 cm height.

Fig. R2 Mechanical experimental data for commercial silicones from *Adv. Funct. Mater.* **2018**, **28**, **1804328**

4. Uncrosslinked chains slightly decrease the modulus, however, they are detrimental for network quality. The achieved “stretchability” of 2300%, i.e. elongation-at-break of 3.3, is lower than a theoretical elongation of ~6, which is expected for a PBA network with the modulus of 0.088 MPa. This suggests non-uniform mesh size distribution.

Our Response: Thanks for your comments.

Firstly, our new elastomers show excellent stretchability due to optimized network structure (Fig. R1a), especially for BAC2 with maximum tensile ratio of 2300%. The elongation-at-break is 24 (not 3.3), much larger than a theoretical elongation of ~6, which is expected for a PBA network with the modulus of 0.088 MPa (the value was provided from you). This exactly suggests uniform mesh size distribution.

Secondly, uncrosslinked chains have very little effect on modulus (Fig. 2f) but increase dielectric constant (Fig. 2e) while their content is no more than 20% (Fig. 2b). Additionally, cyclic actuation test of 50,000 cycles was performed without performance deterioration which demonstrates actuation performance stability in long-term service (Fig. 3f).

5. The idea behind “Selection of number average molecular weight” is unclear. Why does the molecular weight (MW) of crosslinker should match the crosslink density?

Our Response: Thanks for your comments. In this contribution, we innovatively develop a strategy for the optimization of crosslinked polymer network through modulating the dimension of crosslinking agents which can reduce functionality of crosslinking points from 4 to 3 (comparing to small-molecular crosslinker) and thereby increase isotropic homogeneity due to more circle-like meshes (Fig. 1a_{iii} and 1a_{iv}, 1a_{vi}). Additionally, appropriate average molecular weight (dimension) also facilitates the formation of a uniform crosslinked network (Fig. 1a_{iv} and 1a_{vi}). Considering the analysis discussed above, CN9021NS ($\overline{M}_n=28000 \text{ g mol}^{-1}$), whose average molecular weight matches the range of average molecular weight (\overline{M}_c) between crosslinking points in elastomers, was chosen as macromolecular crosslinker for the construction of *n*-butyl acrylic-based elastomer network (BAC series). For comparison, *n*-butyl acrylate homopolymer (BA-S, BA-M and BA-L), crosslinked by the equimolar (taking BAC2 as reference) polyethylene glycol diacrylate (a small molecular crosslinker, $\overline{M}_n=575 \text{ g mol}^{-1}$), CN9893NS (a medium molecular crosslinker, $\overline{M}_n=1600 \text{ g mol}^{-1}$) and CN9014NS (a large molecular crosslinker, $\overline{M}_n=6800 \text{ g mol}^{-1}$), respectively, was also prepared through a similar polymerization process. As a result, among those elastomers, molecular weight of crosslinker for BAC2 (CN9021NS) matches its \overline{M}_c best (Table R3). And BAC2 demonstrates, as expected, the

maximum elongation and ultimate strength. These results demonstrate convincingly that our assumption is feasible. Frankly, the underlying theory and principle need further studying and more experiments to confirm.

Some experiments and data have been added to the **Revised Manuscript** and the **Revised Extended Data**, please refer to them.

Table R3 Comparison of mechanical properties among acrylic dielectric elastomers

	Young's modulus (MPa)	Elongation ($\lambda = L/L_0$)	Ultimate true strength (MPa)	Ultimate true strength/Young's modulus	Toughness (MJ m^{-3})	Density (kg m^{-3})	Estimated \overline{M}_c^* (g mol^{-1})	\overline{M}_n^{**} of crosslinker (g mol^{-1})
VHB™4910	0.211	15.16	14.20	64.57	4.37	1037	35917	--
BA-S	0.112	12.37	7.93	105.72	2.53	1122	73210	575
BA-M	0.147	9.73	9.14	96.21	3.12	1117	55531	$1.6 * 10^3$
BA-L	0.161	9.745	10.39	129.86	3.44	1061	48160	$6.8 * 10^3$
BAC2	0.073	23.97	32.24	366.34	6.77	1008	100911	$2.8 * 10^4$

*: \overline{M}_c means the average molecular weight among crosslinking points, and is estimated from the equation: $\overline{M}_c = 3\rho RT/Y$, where ρ , R , T and Y are density of elastomers, ideal gas constant ($8.314 \text{ J}/(\text{mol} \cdot \text{K})$), Kelvin temperature (293 K) and Young's modulus.

** : Number-average molecular weight of crosslinker (polyethylene glycol diacrylate for BA-S; oligomer CN9893NS for BA-M; oligomer CN9014NS for BA-L and CN9021NS for BAC2). The data is obtained from GPC.

6. As discussed above, the crosslink density of conventional linear chain elastomers has a lower bound due to entanglements, which corresponds to the molecular weight of the entanglements strand M_e . For PBA, $M_e \sim 30,000 \text{ g/mol}$. Is it the reason for the CN9021NS selection?

Our Response: Thanks for your comments. In addition to good compatibility or solubility (dispersion) of crosslinkers in polymer or precursor matrix, appropriate average molecular weight (dimension) which matches the mesh size is main consideration for the CN9021NS selection. For comparison, *n*-butyl acrylate homopolymer (BA-S, BA-M and BA-L), crosslinked by the equimolar (taking BAC2 as reference) polyethylene glycol diacrylate (a small molecular crosslinker, $\overline{M}_n = 575 \text{ g mol}^{-1}$), CN9893NS (a medium molecular crosslinker, $\overline{M}_n = 1600 \text{ g mol}^{-1}$) and CN9014NS (a large molecular

crosslinker, $\overline{M}_n=6800 \text{ g mol}^{-1}$), respectively, was also prepared through a similar polymerization process. As a result, among those elastomers, molecular weight of crosslinker for BAC2 (CN9021NS) matches its \overline{M}_c best (Table R3). And BAC2 demonstrates, as expected, the maximum elongation and ultimate strength. These results demonstrate convincingly that our assumption is feasible. Frankly, the underlying theory and principle need further studying and more experiments to confirm.

We have taken no account of entanglements.

7. In general, network modulus is determined by crosslink density, whereas elongation-at-break also depends on network uniformity. None of these parameters is directly related to the crosslinker MW. It is possible that longer crosslinkers have much lower polymerization rate constant, which promotes network uniformity. In this case, the authors should study mechanical properties as a function of crosslinker MW. Concurrently, one should conduct NMR studies to monitor incorporation of crosslinker to the network structure.

Our Response: Thanks for your professional comments and suggestions. In this contribution, we conceived innovative strategy and then adopted simple methods to prepare new elastomers with excellent performance. Firstly, macro-molecular crosslinker can reduce functionality of crosslinking points from 4 to 3 (comparing to small-molecular crosslinker) and thereby increase isotropic homogeneity due to more circle-like meshes (Fig. 1aiii and 1aiv, 1avi). Secondly, it is beneficial to uniform mesh distribution that molecular weight of crosslinker matches \overline{M}_c of elastomer.

Considering the analysis discussed above, CN9021NS ($\overline{M}_n=28000 \text{ g mol}^{-1}$) was chosen as macromolecular crosslinker for the construction of *n*-butyl acrylic-based elastomer network (BAC series). For comparison, *n*-butyl acrylate homopolymer (BA-S, BA-M and BA-L), crosslinked by the equimolar (taking BAC2 as reference) polyethylene glycol diacrylate (a small molecular crosslinker, $\overline{M}_n=575 \text{ g mol}^{-1}$), CN9893NS (a medium molecular crosslinker, $\overline{M}_n=1600 \text{ g mol}^{-1}$) and CN9014NS (a large molecular crosslinker, $\overline{M}_n=6800 \text{ g mol}^{-1}$), respectively, was also prepared through a similar polymerization process. As a result, among those elastomers, molecular weight of crosslinker for BAC2 (CN9021NS) matches its \overline{M}_c best (Table R3). And BAC2 demonstrates, as expected, the maximum elongation and ultimate strength (Fig. R3). These results demonstrate convincingly that crosslinkers do affect network structure. Some experiments and data have been added to the **Revised Manuscript** and the **Revised Extended Data**, please refer to them.

Fig. R3 Stress-strain curves for VHB™4910, BA-S, BA-M, BA-L and BAC2

As our acrylic elastomers are chemically-crosslinked, liquid NMR cannot be conducted while the abundance of solid-state NMR is too low. In fact, we have tried to test crosslinking density by Magnetic Resonance Crosslink Density Spectrometer. Yet, data error is comparable to testing value due to lack of calibration data for acrylic elastomers.

As you mentioned, polymerization rate is indeed one of likely reasons for the results. Besides, the effect of entanglements and molecular weight distribution of crosslinkers also deserves much attention. Frankly, the underlying theory and principle need further studying and more experiments to confirm. We will continue following this topic. Thanks for your advisable suggestions sincerely.

8. Miscibility of CN9021NS and PBA might be an issue during polymerization reaction. They may phase separate with as the MW increases. In general, CN9021NS and PBA miscibility should be discussed.

Our Response: Thanks for your comments. Firstly, CN9021NS has a good solubility in *n*BA monomer matrix to form homogenous precursor before polymerization. Addition polymerization and curing reacted simultaneously to form uniform crosslinking network. Miscibility of CN9021NS and *n*BA monomer has been thought over when we selected crosslinkers. Secondly, there is only one mechanical loss peak in dynamic mechanical analysis (Fig. R4), and the fracture surface of BAC2 observed by electronic microscope scanning (Fig. R5) demonstrates that our acrylic elastomer is homogenous. Both results testify that there is no phase separation.

Fig. R4 Temperature dependence of mechanical loss for VHB™4910, BAC2, BAC3 and BAC4.

Fig. R5 The fracture surface of BAC2 observed by electronic microscope scanning demonstrates.

9. Gel fraction of the prepared elastomers should be measured. I am surprised about the larger hysteresis in Figure 2h. This suggests poor network quality and significant viscous fraction, which could be due to unreacted species and dangles.

Our Response: Thanks for your comments. Gel fraction has already been measured and shown in Fig. 2b. Compared with VHB™ 4910, hysteresis has been greatly suppressed for BAC series, as demonstrated in Fig. 2g and 2h and Extended Data Fig. 4e. However, hysteresis calculated by stress-strain loops in Fig. 2h is badly dependent on stretch rate. Mechanical loss factor ($\tan \delta$) measured by DMA is more comparable with previous materials. BAC2 ($\tan \delta \sim 0.21$ @ 20°C , 1 Hz) presents much advantages over VHB™ 4910 ($\tan \delta \sim 0.93$ @ 20°C , 1 Hz) and other elastomers ($\tan \delta \sim 0.4$ @ 20°C , 1 Hz for copolymer of acrylate and polyurethane in *Polymer 2018, 149, 39-44*; $\tan \delta \sim 0.4 \sim 1.0$ @ 20°C , 1 Hz for acrylate in *Polymer 2018, 137, 269-275*; $\tan \delta \sim 0.3$ @ 1 Hz for Dow Corning Sylgard 50:1 in *Adv. Funct. Mater. 2018, 28, 1804328*).

Mechanical loss of elastomers with and without uncrosslinked chains has been characterized and then verified that those uncrosslinked chains have little effect on hysteresis (Fig. 2g).

10. The reported dielectric constant is significantly higher than that of PBA. It is unclear how the

uncrosslinked chains “would increase the dielectric constant”. Some dipoles at chain ends are mentioned. However, their concentration is relatively low to make significant contribution to dielectric properties.

Our Response: Thanks for your comments. It deserves some attention and clarification that the dielectric constant of BAC2 is **not significantly higher** than that of PBA.

BA-S, which was cured by small-molecular crosslinker and contained ~98 wt% pBA, can be regarded as PBA. And its measured dielectric constant is 5.36 @ 1 kHz, a little lower than that of BAC2 (5.75 @ 1kHz) crosslinked by 26.4 wt% CN9021NS (Fig. R6a). We attribute this slight difference to those uncrosslinked chains which improve dielectric constant from 5.4 @1 kHz to 5.75 @ 1kHz (Fig. R6b) and the promotion from the flexible long aliphatic polyether backbone inside CN9021NS to movement of chains.

Fig. R6 a) Frequency dependence of dielectric constant and loss for BA-S, BA-M, BA-L and BAC2; b) Comparison of dielectric properties for BAC2 before and after swollen.

11. In line 85, “free volume space of elastomer network” is an odd expression.

Our Response: Thanks for your comments. We have changed this expression to “free volume of elastomers network” in the **Revised Manuscript**.

12. Throughout the manuscript: area strain should be replaced with areal strain.

Our Response: Thanks for your comments. We have consulted an extensive literature, and find that in fact, both expressions have been used broadly. For example, area strain was adopted in *Appl. Phys. Lett.* 2011, 99, 242901; *NPG Asia Mater.* 2019, 11, 62; *Chem. Eng. J.* 2020, 382, 123037; *Adv. Funct. Mater.* 2020, 2008321; *Adv. Funct. Mater.* 2020, 2006639; *Chem. Eng. J.* 2021, 405, 126634, and so on. Areal strain appeared in *Adv. Mater.* 2007, 19, 2218-2223; *Extreme Mechanics Letters* 2020, 38, 100752, and so on.

We feel these changes fully address the Reviewers' comments. Again, we thank the referees for the appreciation for this work and also for the valuable remarks. Thank you for your consideration and help with this manuscript. Please feel free to contact me if any additional information is needed for your decision.

Sincerely,

Zhi-Min Dang

REVIEWERS' COMMENTS

Reviewer #1 (Remarks to the Author):

Thank you for your detailed response to the reviewer comments. This has taken a lot of effort, but also made the manuscript much stronger and more rigorous. The paper conclusions are justified, and the limitations to the work more clearly stated (for example, the comparison with VHB does not fully explore the ultra large strain range that has been achieved by pre-stretch, but this has been put into context). The improved frequency response and strain to field are valuable contributions, as is the publication of the general materials approach conveyed by the authors.

John D Madden

Reviewer #2 (Remarks to the Author):

The authors have addressed my comments carefully and most of my concerns have been clarified. One question raised in Comment #2 was still not answered satisfactorily. The comparison made between the authors' results and those in literature was set by an artificial standard rather than from a point of view of applications of DEAs. It is suggested the authors focus on discussing the advantages of material parameters but weaken the statement of the comparison with VHB. Overall, the revised manuscript is acceptable without need of further review from my side.

Reviewer #3 (Remarks to the Author):

I am still not convinced about the novelty about this paper. I see neither concept nor design strategy behind the prepared materials. It is unclear what fundamental issue is addressed and resolved in this study. The paper is narrowly focused on improving DEA performance of the commercial and rather antique VHB material. Yet, in recent years, many new materials have been developed with much better DEA performance than VHB. Table R1 outlines many improvements with respect to VHB. Some of them such as rapid response are significant and deserves reporting in an applied journal. But many of the outlined improvements look incremental. Big issues such as pre-strain requirement remains unresolved. Like VHB, the prepared materials require pre-strain to eliminate electromechanical instability because of weak strain-hardening. There are many contemporary materials that allow >100% actuation without pre-strain.

In their reply, the authors make misleading statements and use odd terminology, which cause significant doubts about credibility of this study and authors experience in the field. Here a few examples of such statements:

“Soft materials bring about troubles” such as electric breakdown and output force. There is no physical relation between softness and breakdown. Also, the output force depends on applied voltage. It is true, that softer materials require lower field for actuation (without external load). But, if an external load is applied, a higher field should be applied to move the load.

A related statement: “Softness will deteriorate insulating property badly.” As mentioned above: There is no physical relation between softness and breakdown. Transformer oil is very soft, while

being one of the best insulators.

“If there are only physical entanglements among poly(n-butyl acrylate) chains, the slippage appears under large deformation. Hence, the theoretical minimum modulus based on entanglements is unpractical.” There is a fundamental misunderstanding of polymer science in this statement. In covalently crosslinked polymer networks, chain entanglements behave as permanent crosslinks that set a lower limit for modulus. There are a lot of practical implications for soft materials design associated with entanglements. Lowering the entanglement density is a pathway towards soft materials.

The modulus measurements depend on sample size and shape. Modulus is a material property (like density) which should not depend on sample dimensions. It may depend on strain rate and temperature, but not on sample dimensions.

Comments from reviewers:

Reviewer #1: Thank you for your detailed response to the reviewer comments. This has taken a lot of effort, but also made the manuscript much stronger and more rigorous. The paper conclusions are justified, and the limitations to the work more clearly stated (for example, the comparison with VHB does not fully explore the ultra large strain range that has been achieved by pre-stretch, but this has been put into context). The improved frequency response and strain to field are valuable contributions, as is the publication of the general materials approach conveyed by the authors.

Our Response: Thanks for your recognition for our work in earnest. Your professional comments are valuable for us to promote our manuscript indeed.

Reviewer #2: The authors have addressed my comments carefully and most of my concerns have been clarified. One question raised in Comment #2 was still not answered satisfactorily. The comparison made between the authors' results and those in literature was set by an artificial standard rather from a point view of applications of DEAs. It is suggested the authors focus on discussing the advantages of material parameters but weaken the statement of the comparison with VHB. Overall, the revised manuscript is acceptable without need of further review from my side.

Our Response: Thanks for your appreciation on our work and reasonable comments. As regard to your concerns, we would like to give further explanation.

Firstly, in comment #2, the reviewer considered that the actuation strain of VHB achieved by us was too small compared to those in literature, and thereby the actuation performance of our new materials was not as good as VHB. And we have ascribed the larger actuation strain of VHB in literature to two reasons: much higher actuation electric field and elaborate techniques or ingenious structure. For example, Pelrine et al used VHB 4910 to obtain 158% circular strain under 412 MV/m and 215% linear strain under 239 MV/m with pre-strain, respectively (*Science* **2000**, **287**, **836-839**). Huang et al has achieved 488% areal strain under 5.25 kV (the value was drawn from figure) with dead load. The equal-biaxial dead load has been proved to suppress electric breakdown and consequently enables elastomers to gain a greater voltage-induced deformation than rigid constraint does (*Appl. Phys. Lett* **2012**, **100**, **041911**). However, in our work, the maximum actuation electric field is merely 70 MV/m and testing apparatus has been simplified to display elastomer performance only. In fact, approximately 30% area strain @70 MV/m for VHBTM 4910 obtained here is not too small compared with those in literature under the same electric field (*Appl. Phys. Lett* **2012**, **100**, **041911**; *Soft Matter* **2012**, **8**, **6167-6173**). Additionally, the commonly-used

maximum area actuation strain for VHBTM4910 is 7.5% @ 17 MV/m without pre-strain, which was reported by Spontak (*Macromol. Rapid Commun.* 2007, 28, 1142–1147; *Polymer* 2018, 137, 269-275; *Chem. Eng. J.* 2020, 382, 123037). Then, taking this value as reference, area strain (18.5% @ 15 MV/m without pre-strain, 118% @ 70 MV/m with equiaxial pre-strain) for BAC2 still has its advantages.

Here, we would like to supplement further illustration to this concern. On the one hand, actuation strain of dielectric elastomer when used as dielectric elastomer actuator is generally no more than 50% (*Nature* 2021, 591, 66-71; *Sci. Robot.* 2019, 4, eaaz6451; *Nature* 2019, 575, 324-329). Therefore, larger strain at lower electric field is much more important for applications rather than huge strain at extreme conditions, which has received more attention from researchers majored in solid mechanics and electromechanical instability (*Science* 2000, 287, 836-839; *Appl. Phys. Lett* 2012, 100, 041911). On the other hand, in addition to actuation strain, response speed and strain drift are other essential indicators for actuation performance (*Adv. Funct. Mater.* 2015, 25, 1656–1665; *Adv. Funct. Mater.* 2018, 28, 1804328; *Nature* 2019, 575, 324-329; *IEEE/ASME Transactions on Mechatronics* 2019, 24, 36-44). And BAC2 features much wider frequency bandwidth (>100 Hz for BAC2 and <10Hz for VHB) and greatly suppressed strain drift.

In conclusion, despite smaller maximum strain compared with that of VHB in extreme circumstances, we have every reason to claim that our new elastomer BAC2 demonstrates far better actuation performance than VHB due to its larger actuation strain under lower electric field (118% @70 MV/m and 18.5% @15 MV/m), faster response, wider frequency bandwidth (>100 Hz).

Secondly, in order to maximize the influence of our work and meanwhile draw the attention of soft actuator researchers, VHB, the most-widely used elastomer, was taken as a reference here. However, as VHB is a commercial elastomer, we cannot obtain its precise components and structure. Therefore, we focus on discussing the advantages of material parameters as well as the relationship between structure and properties but weaken the statement of the comparison with VHB.

Last, we would like to thank you again for your recognition and important suggestions.

Reviewer #3: I am still not convinced about the novelty about this paper. I see neither concept nor design strategy behind the prepared materials. It is unclear what fundamental issue is addressed and resolved in this study. The paper is narrowly focused on improving DEA performance of the commercial and rather antique VHB material. Yet, in recent years, many new materials have been developed with much better DEA performance than VHB. Table R1 outlines many improvements

with respect to VHB. Some of them such as rapid response are significant and deserves reporting in an applied journal. But many of the outlined improvements look incremental. Big issues such as pre-strain requirement remains unresolved. Like VHB, the prepared materials require pre-strain to eliminate electromechanical instability because of weak strain-hardening. There are many contemporary materials that allow >100% actuation without pre-strain.

Our Response: Thanks for your comments. Here, we are going to respond to your concerns point by point carefully. But at the very beginning, we would like to reassert the novelty about this paper as clearly as possible. In this contribution, we innovatively select macro-molecular crosslinker to construct network to realize significant improvement of the whole performance, which exactly is revolutionary. And it's worth noting that the influence of the molecular weight to network structure and even elastomer properties has been investigated deeply. As a result, a new polyacrylate dielectric elastomer with optimized polymer network was synthesized by employing difunctional macromolecular crosslinking agent whose molecular weight matches the average molecular weight among crosslinking points (\overline{M}_c) of elastomers. The synthesized acrylic elastomer features desirable Young's modulus (~0.073 MPa), high elongation (~2400%) and low mechanical loss ($\tan \delta_m=0.21$ @1 Hz, 20 °C), satisfactory dielectric properties ($\epsilon_r=5.75$, $\tan \delta_e=0.0019$ @1 kHz). Accordingly, large actuation strain (118% @ nominal electric field of 70 MV m⁻¹), high energy density (0.24 MJ m⁻³ @ nominal electric field of 70 MV m⁻¹) and rapid response (bandwidth above 100 Hz) are successfully achieved in this elaborate polyacrylate elastomer. Besides, it is a huge challenge to simultaneously improve the softness and toughness of elastomer and such a paradox is solved in this contribution through the optimization of crosslinked network. The BAC2 sample not only exhibits a low Young's modulus (0.073 MPa) and a high elongation (~2400%) but also displays a high toughness (6.77 MJ m⁻³) and a high strength (32.23 MPa). The excellent mechanical property displayed here is seldom observed in previously reported works. Conventionally, most strategies with an effort of contributing to one performance improvement (i.e., low modulus, high dielectric constant, low mechanical loss) for acrylic elastomers, may cause the degradation of the others (*J. Mater. Chem. C* 2019, 7, 12139-12150; *J. Mater. Chem. C* 2018, 6, 2043-2053; *Adv. Eng. Mater.* 2019, 21, 1900481). The design and preparation of high-quality dielectric elastomers remains a challenging problem for the mass application of DEAs. However, the material design strategy we have proposed in this work realizes collaborative improvement of many performances.

Next, in this paper, we focus on verifying design concept and discussing the relationship between

structure and properties but weaken the statement of the comparison with VHB, as its precise components and structure is unavailable. And VHB, the most-widely used elastomer, was just taken as a reference to maximize the influence of our work and meanwhile draw the attention of soft actuator researchers (Table R1). Therefore, this paper is not narrowly focused on improving DEA performance of the commercial VHB material.

Thirdly, we must admit that the pre-strain requirement has not been resolved in this work due to weak strain-hardening of BAC2, like VHB. However, pre-strain is not the focus of this paper, and actuation strain of BAC2 without pre-strain (18.5% @ 15 MV/m) can satisfy most applications (*Nature* 2021, 591, 66-71; *Sci. Robot.* 2019, 4, eaaz6451; *Nature* 2019, 575, 324-329). Then, we have reviewed more than 250 papers published since 2000, especially since 2010 (*Macromol. Rapid Commun.* 2010, 31, 10–36; *Prog. Polym. Sci.* 2015, 51, 188-211; *Extreme Mechanics Letters* 2020, 38, 100752). And very few works which reported >100% actuation strain without pre-strain have been found. The most famous series works were interpenetrating polymer networks (IPNs) finished by Pei's group (*Adv. Mater.* 2006, 18, 887-891; *Smart Mater. Struct.* 2007, 16, S280-S287; *Proc. SPIE* 2008, 6927, 69272C). The film is first pre-strained, and then a multifunctional monomer additive is sprayed onto the host film and polymerized forming an interpenetrating polymer network. Upon releasing the film it retains most of the applied pre-strain, with the additive network being in compression and the host film in tension (Fig. R1). And Pei et al has reported 233% area strain for IPN (VHB 4910-HDDA) (*Adv. Mater.* 2006, 18, 887-891; *Smart Mater. Struct.* 2007, 16, S280-S287), 146% area strain for IPN (VHB 4905-TMPTMA) and 300% area strain for (VHB 4910-TMPTMA) (*Proc. SPIE* 2008, 6927, 69272C) without external pre-strain. Subsequently, they reported a new elastomer capable of high actuation strain (>100%) without pre-strain by adjusting the content of crosslinker and plasticizer (*J. Polym. Sci., Part B: Polym. Phys.* 2013, 51, 197–206). In fact, the actuation testing conditions used by Pei et al was a little difference to those adopted by us. For example, they employed a bias air pressure to control deformation direction, which, in the meantime, deformed the elastomer a little. Besides, they retained annular passive part between rigid frame and electrode-coated elastomer (active part), and this passive part was stretched simultaneously when elastomer was actuated (Fig. R2). These ingenious and delicate setting may contribute to large actuation strain. In 2012, Spontak et al. reported a dielectric gel (*Adv. Funct. Mater.* 2012, 22, 2100–2113). The thermoplastic elastomer gels (TPEGs) composed of styrenic triblock copolymers swollen with a midblock-selective solvent

exhibits remarkable electromechanical properties as high-performance dielectric elastomers. When the copolymer concentration is 45%wt, the maximum actuated area strain reaches ~115% without pre-strain. The last work is about bottlebrush elastomers. The bottlebrush elastomers composed of multiple, covalently-linked side chains along the network strands which acted as solvent exhibited 300% area strain without pre-strain (*Adv. Mater.* 2016, 29, 1604209).

However, the maximum area strain of majority of literature is below 20% without pre-strain (*ACS Appl. Mater. Interfaces* 2020, 12, 23432–23442; *PNAS* 2019, 116, 2476-2481; *Adv. Funct. Mater.* 2018, 28, 1804328; *Polymer* 2018, 149, 39e44; *Nature* 2019, 575, 324-329; *NPG Asia Mater.* 2019, 11, 62; *Chem. Eng. J.* 2021, 405, 126634; *Nature* 2021, 591, 66-71). In addition to actuation strain, response speed and strain drift are other essential indicators for actuation performance and attract increasing attention of researchers (*Adv. Funct. Mater.* 2015, 25, 1656–1665; *Adv. Funct. Mater.* 2018, 28, 1804328; *Nature* 2019, 575, 324-329; *IEEE/ASME Transactions on Mechatronics* 2019, 24, 36-44). In conclusion, despite maximum strain seemed not prominent, we have every reason to claim that our new elastomer BAC2 features much desirable actuation performance due to its large actuation strain under low electric field (118% @70 MV/m and 18.5% @15 MV/m), fast response, wide frequency bandwidth (>100 Hz) as well as suppressed strain drift.

Table R1 Comparison of performances between VHBTM4910 and BAC2

	VHB TM 4910	BAC2
Dielectric constant @ 1kHz	4.4	5.75
Dielectric loss @ 1kHz	0.0229	0.002
Young's modulus at 5% strain (MPa)	0.211	0.073
Mechanical loss factor (tan δ_m) @20°C, 1 Hz	0.21	0.93
Actuation sensitivity (MPa ⁻¹)	20.995	78.8
Elongation at break (%)	~1500	~2400
Ultimate true strength (MPa)	14.2	32.23
Ultimate true strength to modulus	67.32	441.53
Toughness (MJ m ⁻³)	4.37	6.77
Area strain @ 15 kV m ⁻¹ without pre-strain	4.5%	18.5%
Area strain @ 70 kV m ⁻¹ with pre-strain	34%	118%
Estimated energy density (MJ m ⁻³)	0.042	0.242
Frequency bandwidth (Hz)	<10	>100

Fig. R1 Fabrication steps of IPN elastomer films (*Proc. SPIE 2007, 6524, 652408*).

Fig. R2 (a) Pictures of the actuated elastomer films. (b) Electromechanical strain versus applied electric field relationships of elastomers with different crosslinker concentrations. (*J. Polym. Sci.,*

Part B: Polym. Phys. 2013, 51, 197–206)

1. In their reply, the authors make misleading statements and use odd terminology, which cause significant doubts about credibility of this study and authors experience in the field. Here a few examples of such statements.

Our Response: Thanks for your comments. Hereinafter, we have made a careful response to your

comments point to point.

2. “Soft materials bring about troubles” such as electric breakdown and output force. There is no physical relation between softness and breakdown. Also, the output force depends on applied voltage. It is true, that softer materials require lower field for actuation (without external load). But, if an external load is applied, a higher field should be applied to move the load.

Our Response: Thanks for your comment.

Firstly, regardless of testing conditions, the short-time breakdown of a solid dielectric is generally attributed to two mechanisms: intrinsic electric breakdown and electromechanical breakdown (*IEEE Trans. Power Electron.* 1992, 7, 251-257; *Adv. Funct. Mater.* 2012, 22, 2100–2113; *Prog. Mater. Sci.* 2019, 100, 187-225). For very short electrical stress duration, and carefully controlled conditions as purity, homogeneity, and temperature of the material, the electric strength increases to an upper limit, called intrinsic electric strength. Because times to failure can be as short as 10 ns, most concepts of intrinsic breakdown are electronic in nature (*IEEE Trans. Electr. Insul.* 1980, 15, 206-224). At fields approaching breakdown, electrostatic compressive forces are numerically comparable to Young’s modulus of materials and will cause deformation of materials such as polymers or fracture of materials such as ceramics (*IEEE Trans. Power Electron.* 1992, 7, 251-257). Specially, failure of elastomers usually occurs by electromechanical instability due to their excellent softness (<1 MPa). There is an empirical equation for evaluating electromechanical breakdown (E_{em}) which is called Stark-Garton model (*Nature* 1955, 176, 1225-1226):

$$E_{em} = 0.6 \left(\frac{Y}{\epsilon_0 \epsilon_r} \right)^{\frac{1}{2}}$$

where ϵ_0 , ϵ_r , and Y are the dielectric constant of vacuum, dielectric constant of elastomer, and Young’s modulus of elastomer, respectively. Hence, soft materials usually withstand lower electric fields. However, mechanics of solid breakdown is still not completely clear up to now and many conclusions and claims are semi-empirical. We will continue to follow relative studies and deepen our understanding in the future.

Secondly, at electric fields, the Maxwell stress (P_e) (*Science* 2000, 287, 836-839) in the thickness direction can be defined by $P_e = \epsilon_0 \epsilon_r E^2$, where E is the applied electric field. Thus, a dielectric elastomer will expand as shown in Fig. R3(a) and R3(b). For an incompressible material, a state of biaxial stress can cause the same deformation as a state of uniaxial stress dose when $t_2 = t_3 = P_e$ (Fig. R3(b) and R3(c)). Hence, the maximum output force is decided by breakdown strength. If the

load is larger than the maximum output force, elastomers cannot move the load but may wrinkle. As we have stated above that softness may bring about low electric breakdown, the maximum output force of soft materials is gentle. Besides, soft materials are weak in mechanical support and tend to deform out of plane such as wrapping or wrinkling, when a large external load is applied. In summary, the output force depends on applied voltage but the maximum output force is closely related to elastic modulus. Our original expression in response letter is not enough rigorous and scientific, and we will pay attention to it in the future.

Fig. R3 A dielectric elastomer in reference state (a), electrically-actuated state (b) and equivalent thermodynamic state (c).

3. A related statement: "Softness will deteriorate insulating property badly." As mentioned above: There is no physical relation between softness and breakdown. Transformer oil is very soft, while being one of the best insulators.

Our Response: Thanks for your comments. The semi-empirical relation between softness and breakdown has been thoroughly illustrated above. Here, we would like to make an explanation on the statement that "Transformer oil is very soft, while being one of the best insulators". Table R2 listed dielectric constants and breakdown strengths of selected insulators especially including transformer oil, silicone oil and air. It deserves noting that the unit of breakdown strength in Table R2 is different from commonly-used unit now, and $1 \text{ MV/m} = 25.4 \text{ V/mil}$. Hence, the breakdown strength of transformer oil is about 12-20 MV/m, which is lower than that of BAC2 (23.4 MV/m). The extensive use of transformer oil as liquid insulator should be attributed to higher breakdown strength than air, high fire point and flash point, suitable viscosity and low density. Nevertheless, it is somewhat inappropriate to claim transformer oil as one of the best insulators.

Table R2 Dielectric constants (K) and breakdown strengths (V_b) of selected insulators (*IEEE Trans.*

Power Electron. 1992, 7, 251-257)

Insulator	K	$V_b(V/mil)$
Air	1.000585	75
Aluminum oxide	7.0–10	250–380
Bakelite (general purpose)	6.0	300
Castor oil	4.5	350
Ceramics	5.5–7.5	200–350
Ethylene glycol	39	500
High-voltage ceramic (barium titanate composite and filler)	500–6 000	50
Kapton (polyimide)	3.6	7 000
Kraft paper (impregnated)	6.0	2 000
Lucite	3.3	500
Mylar	3.25	7 000
Paraffin	2.25	250
Polycarbonate	2.9	7 000
Polyethylene	2.2	4 500
Polypropylene	2.2	7 500
Polystyrene	2.5	600
Polysulfone	3.1	8 000
Pyrex glass	4.6	500
Quartz, fused	3.85	500
Reconstituted mica	7.8	1 600
Silicone oil	2.8	350
Sulfur hexafluoride	1.0	200
Sulfur	4.0	
Tantalum oxide	27	13 000 ^a
Teflon	2.0	2 200–4 400
Titanium dioxide ceramics	15–500	
Transformer oil	2.2	300–500
Water	80	500 ^b

4. *"If there are only physical entanglements among poly(n-butyl acrylate) chains, the slippage appears under large deformation. Hence, the theoretical minimum modulus based on entanglements is unpractical."* There is a fundamental misunderstanding of polymer science in this statement. In covalently crosslinked polymer networks, chain entanglements behave as permanent crosslinks that set a lower limit for modulus. There are a lot of practical implications for soft materials design associated with entanglements. Lowering the entanglement density is a pathway towards soft materials.

Our Response: Thanks for your comments. Here we would like to make some clarification to our statement appeared in last response letter. This statement is only aimed at network of poly(n-butyl acrylate) chains. According to our experimental results, when there are only physical entanglements without any chemical crosslinking, poly(n-butyl acrylate) chains are in a viscous-flow state at room temperature and cannot retain their shape due to excellent chain flexibility. Therefore, the slippage

among poly(*n*-butyl acrylate) chains appears under large strain. Hence, we consider that covalent crosslinking plays a critical role in optimized polymer network rather than physical entanglements. We will pay more attention to physical entanglements in following researches. Thanks for your comments again.

5. The modulus measurements depend on sample size and shape. Modulus is a material property (like density) which should not depend on sample dimensions. It may depend on strain rate and temperature, but not on sample dimensions.

Our Response: Thanks for your comments. As you said, modulus is exactly a material property which should not depend on sample dimensions. In fact, for the sake of accuracy and stability of measurements, specific shape and size of samples and corresponding strain rate are recommended by ISO 37-2005.

We feel these changes fully address the comments from Editor and Reviewers. Again, we thank the referees for the appreciation for this work and for the valuable remarks. Thank you for your consideration and help with this manuscript. Please feel free to contact me if any additional information is needed for your decision.

Sincerely,

Zhi-Min Dang